# The Extract of *Periplaneta americana* (L.) Promotes Hair Regrowth in Mice with Alopecia by Regulating the FOXO/PI3K/AKT Signaling Pathway and Skin Microbiota

**DOI:** 10.3390/cimb47080619

**Published:** 2025-08-04

**Authors:** Tangfei Guan, Xin Yang, Canhui Hong, Zehao Zhang, Peiyun Xiao, Yongshou Yang, Chenggui Zhang, Zhengchun He

**Affiliations:** 1Yunnan Provincial Key Laboratory of Entomological Biopharmaceutical R&D, College of Pharmacy, Dali University, Dali 671000, China; 2National-Local Joint Engineering Research Center of Entomoceutics, Dali 671000, China; 3West China School of Public Health and West China Fourth Hospital, Sichuan University, Chengdu 610041, China

**Keywords:** alopecia, *Periplaneta americana* (L.), network pharmacology, transcriptomics, metabolomics, skin microbiota

## Abstract

Alopecia, a prevalent dermatological disorder affecting over half of the global population, is strongly associated with psychological distress. Extracts from *Periplaneta americana* (L. PA), a medicinal insect resource, exhibit pharmacological activities (e.g., antioxidant, anti-inflammatory, microcirculation improvement) that align with core therapeutic targets for alopecia. This study aimed to systematically investigate the efficacy and mechanisms of PA extracts in promoting hair regeneration. A strategy combining network pharmacology prediction and in vivo experiments was adopted. The efficacy of a Periplaneta americana extract was validated by evaluating hair regrowth status and skin pathological staining in C57BL/6J mice. Transcriptomics, metabolomics, RT-qPCR, and 16s rRNA techniques were integrated to dissect the underlying mechanisms of its hair-growth-promoting effects. PA-011 significantly promoted hair regeneration in depilated mice via multiple mechanisms: enhanced skin superoxide dismutase activity and upregulated vascular endothelial growth factor expression; modulated FOXO/PI3K/AKT signaling pathway and restored skin microbiota homeostasis; and accelerated transition of hair follicles from the telogen to anagen phase. PA-011 exerts hair-promoting effects through synergistic modulation of FOXO/PI3K/AKT signaling and the skin microbiome. As a novel therapeutic candidate, it warrants further systematic investigation for clinical translation.

## 1. Introduction

Alopecia, a common skin disorder characterized by hair follicle atrophy, impairs physical appearance and exacerbates psychological distress and social dysfunction [1]. Globally, ~40% of women and 85% of men are affected [2]. Current U.S. FDA-approved treatments (finasteride and minoxidil) exhibit limited long-term use due to high costs and adverse effects [3,4], underscoring the urgent need for safe, effective alternatives in clinical and cosmetic applications.

Ap pathogenesis involves multifactorial mechanisms, including physical trauma, chemotherapy, androgen excess, sebum dysregulation, and growth factor imbalance [5]. Inflammation and aberrant signaling pathways are central drivers [6]. For example, vascular endothelial growth factor (VEGF) promotes hair growth by enhancing HF size and hair shaft thickness [7], while superoxide dismutase (SOD)—an antioxidant critical for counteracting oxidative stress—mitigates follicular damage [8]. Oxidative stress induces apoptosis in anagen-phase follicles, disrupting matrix keratinocyte proliferation and causing hair loss [9]. Emerging therapies target these mechanisms: amygdalin inhibits JAK2/STAT3-driven inflammation in Ap mice [10]; ginkgo polysaccharides reduce inflammation [11]; Rosa rugosa water extract enhances VEGF expression to accelerate regrowth [12]; and NMN alleviates oxidative stress to promote follicular cell proliferation [13]. Collectively, strategies inhibiting cutaneous inflammation/oxidative stress, delaying apoptosis, and upregulating VEGF represent validated therapeutic targets for Ap.

Network pharmacology, grounded in the “disease-gene-target-drug” interaction network, systematically elucidates the synergistic effects of compounds and the potential mechanisms of multi-component, multi-target drugs at the molecular level [14]. Studies on mRNA, protein, and metabolite variations in alopecia patients have deepened our understanding of its pathogenesis [15]. Mining disease biomarkers and targets through multi-omics approaches represents a current research frontier, enabling comprehensive dissection of alopecia mechanisms, screening of therapeutic targets, and formulation of personalized strategies.

Skin diseases (e.g., atopic dermatitis, acne, psoriasis, and alopecia) represent the fourth most prevalent human disease category, affecting ~one-third of the global population [16]. The human skin microbiome, comprising millions of bacteria, fungi, and viruses [17], is increasingly recognized for its role in immune-mediated inflammatory skin conditions [18] and maintenance of skin/hair health [19,20]. It regulates physiological homeostasis and protects against pathogen invasion [21]. Emerging evidence links scalp microbiome dysbiosis to scalp disorders (e.g., allergies, alopecia) [22], yet mechanisms underlying microbiome–alopecia interactions remain poorly understood. Focusing on the scalp microenvironment for hair growth, we hypothesize that compositional and functional shifts in the scalp microbiome influence alopecia pathogenesis by regulating local microecology [23]. Most prior studies on scalp microbiomes have characterized compositional changes associated with inflammation/disease but lacked mechanistic insights into functional regulation. Whether microbiome regulation can promote hair regrowth remains a highly valuable direction for exploration.

*Periplaneta americana* (L. PA), a traditional medicinal insect in China, contains abundant bioactive components. Modern pharmacological studies have demonstrated that PA extracts exhibit multiple beneficial effects, including antioxidant [24], anti-inflammatory [25], antibacterial [26], antifibrotic [27], and tissue repair-promoting activities [28]. A preparation with PA as the main component, “Kangfuxin Liquid,” has been produced and widely circulated in the market, serving as a clinically commonly used drug for skin trauma [29] and tissue repair promotion [30]. These bioactivities are highly congruent with the therapeutic objectives for Ap.

Building on the preliminary findings of our research group, this study identified that PA-011 (an extract of PA, named according to the characteristics of the extraction process in Section 2.2) effectively promotes skin tissue repair and significantly enhances hair growth in damaged skin areas. Therefore, this study aims to investigate the hair-growth-promoting effects and mechanisms of the PA extract (PA-011) through a combination of network pharmacology prediction, in vivo animal experiments, and multi-omics integration.

## 2. Materials and Methods

### 2.1. Major Reagents and Instruments

Depilatory cream (Veet, Beijing, China, 202346); Minoxidil (Beijing Nuokai Technology Co., Ltd., Beijing, China, KYG1936); 4% paraformaldehyde (Servicebio, Wuhan, China, 202403); Isoflurane (Shandong Animal Husbandry Technology Co., Ltd., Jinan, China, 202404); Absolute ethanol (Sinopharm Chemical Reagent Co., Ltd., Shanghai, China, 100092683); HE staining kit (Servicebio, Wuhan, China G1003); Tissue dehydrator (DIAPATH, Donatello, Jinan, China, ); Tissue embedding machine (Wuhan Junjie Electronics Co., Ltd., Wuhan, China, JB-P5); Pathological microtome (Leica Microsystems Shanghai Co., Ltd., Shanghai, China, RM2016); Tissue flotation bath (Zhejiang Jinhua Kedi Instrument Equipment Co., Ltd., Jinhua, China, KD-P); Upright optical microscope (Nikon, Tokyo, Japan, NIKON ECLIPSE E100); Imaging system (Nikon, Japan, NIKON DS-U3); Analytical balance (Mettler-Toledo Instruments (Shanghai) Co., Ltd., Shanghai, China, ME203E/02); Three-button electronic digital caliper (Guilin Guanglu Digital Measurement & Control Co., Ltd., Guilin, China, SF2000); Tissue grinder (MeiBi, Jiaxing, China, MB-96).

### 2.2. Extraction Method of PA Extract PA-011

Five kilograms of dried PA bodies (Yunnan Jingxin Biotechnology Co., Ltd., Dali, China) were pulverized and extracted via cold maceration with 4-fold weight of 95% ethanol. The extract was concentrated and defatted to obtain a defatted extract. The defatted extract was dissolved in 6-fold weight of 45% aqueous ethanol, stored at 4 °C overnight, filtered to remove insoluble matter, and the filtrate was concentrated under reduced pressure to a small volume before freeze-drying, yielding PA extract PA-011.

### 2.3. Network Pharmacology Analysis of PA-011 in Ap

#### 2.3.1. Identification of PA-011 Components

A precise amount of PA-011 sample was weighed and mixed with 600 µL of methanol solution containing 4 ppm 2-chloro-L-phenylalanine, followed by vortexing for 30 s. The mixture was homogenized at 50 Hz for 120 s, sonicated at room temperature for 10 min, and centrifuged at 12,000 rpm for 10 min at 4 °C. The supernatant was filtered through a 0.22 µm membrane. Separation was performed using a Thermo Vanquish ultra-high performance liquid chromatography (UHPLC) system equipped with an ACQUITY UPLC^®^ HSS T3 column (2.1 × 100 mm, 1.8 µm). The mobile phase flow rate was 0.3 mL/min, column temperature was maintained at 40 °C, and injection volume was 2 µL. For positive ion mode, the mobile phase consisted of 0.1% formic acid in acetonitrile (B2) and 0.1% formic acid in water (A2). For negative ion mode, the mobile phase was acetonitrile (B3) and 5 mM ammonium formate in water (A3). Data were acquired using a Thermo Q Exactive mass spectrometer (ESI source) in both positive and negative ion modes. The positive ion spray voltage was 3.50 kV, and the negative ion voltage was −2.50 kV. Sheath gas and auxiliary gas flows were set to 40 arb and 10 arb, respectively, with a capillary temperature of 325 °C. Full-scan MS1 was performed at a resolution of 70,000 over the *m*/*z* range of 100–1000. Higher-energy collisional dissociation (HCD) was applied for MS2 fragmentation with a collision energy of 30 eV, a resolution of 17,500, and dynamic exclusion of MS/MS information for the top 10 ions. PA-011 lyophilized powder was dissolved in 10 µL of Solution A, centrifuged at 14,000× *g* for 20 min at 4 °C, and 1 µg of the supernatant was injected into a Q Exactive HF-X mass spectrometer (Nanospray Flex™ source). The ion spray voltage was 2.4 kV, and the transfer tube temperature was 275 °C. Data-dependent acquisition (DDA) was performed with full-scan MS1 at a resolution of 120,000 (200 *m*/*z*) over *m*/*z* 100–1500, using an AGC target of 3 × 10^6^ and a maximum injection time of 80 ms. The top 40 precursor ions were fragmented via HCD with a collision energy of 27%, a resolution of 15,000 (200 *m*/*z*), an AGC target of 5 × 10^4^, and a maximum injection time of 45 ms. Raw data were processed using de novo analysis to determine peptide sequences.

#### 2.3.2. Target Acquisition for PA-011

Small-molecule compounds identified in PA-011 were retrieved from the PubChem database (https://pubchem.ncbi.nlm.nih.gov/, accessed on 23 May 2024) for information and structures, then input into the SwissADME database (http://www.swissadme.ch/index.php, accessed on 23 May 2024) for screening. Only compounds with therapeutic disease targets and a probability > 0.01 were retained for target information. For peptide sequences, the top 100 sequences with the highest −10 lgP values from peptide analysis and sequences with a confidence score > 95% from novel peptide analysis were integrated and input into the Emboss database (https://www.ebi.ac.uk/Tools/seqstats/emboss_pepstats/, accessed on 23 May 2024) to obtain protein information. Based on the characteristics of food-derived anti-inflammatory and neuroprotective peptides, sequences with a charge value > 0 were selected, yielding 129 sequences. The 129 peptide sequences were converted to SMILES format using the Nover Por web tool (https://www.novoprolabs.com/tools/convert-peptide-to-smiles-string, accessed on 23 May 2024), then input into the Similarity Ensemble Approach (SEA) database (https://sea.bkslab.org/, accessed on 23 May 2024) for target protein prediction. Target protein information from both small molecules and peptides was integrated and deduplicated, resulting in 474 unique target protein entries.

#### 2.3.3. Acquisition of Ap Disease Targets

Using “Ap” as the search term, disease gene data were retrieved from the OMIM (http://www.ncbi.nlm.nih.gov/omim, accessed on 28 May 2024), GeneCards (http://www.genecards.org, accessed on 28 May 2024), and DisGeNET (https://www.disgenet.org/search, accessed on 28 May 2024) databases as Excel files. The search results from the three databases were merged, duplicate disease target information was removed, and the disease targets and peptide-predicted targets were imported into the Venn 2.1.0 tool to identify intersecting gene information for Ap across peptides.

#### 2.3.4. Construction of Protein–Protein Interaction (PPI) Network for Disease Targets

The intersecting disease targets were input into the STRING database (https://string-db.org/, accessed on 30 May 2024), a protein–protein interaction retrieval and prediction platform. Nodes without connections in the network were hidden, and remaining parameters were set to default values before exporting relevant data. The protein network was analyzed using Cytoscape 3.9.1 software with the Centiscape 2.2 plugin, applying default screening criteria: degree (>10.5854), closeness (>0.0130095), and betweenness (>40.73171).

#### 2.3.5. GO and KEGG Enrichment Analysis

Intersecting disease target information was imported into the DAVID database (https://david.ncifcrf.gov/, accessed on 30 May 2024), where OFFICIAL_GENE_SYMBOL was selected for gene ontology (GO) biological process analysis and Kyoto Encyclopedia of Genes and Genomes (KEGG) pathway enrichment analysis. Enrichment results were visualized using the online bioinformatics platform (http://www.bioinformatics.com.cn/, accessed on 30 May 2024).

#### 2.3.6. Construction of Component-Target-Pathway Network

A network working table was created using the obtained peptide target information, disease intersection targets, and KEGG pathway target data. The integrated data were formatted into a tab-delimited file and imported into Cytoscape 3.9.1 to construct a component-target-pathway network diagram. Node connectivity (degree) was calculated using the Centiscape 2.2 plugin in Cytoscape 3.9.1 to evaluate the interactions among drug peptides, therapeutic targets, and disease pathways.

#### 2.3.7. Molecular Docking

To validate the binding activity between core target proteins and corresponding active components, molecular docking was performed using AutoDock Vina software (v1.1.2). The docking targets were the most highly connected target proteins selected from the PPI network and their corresponding active components. Structural files of target proteins and active components were obtained from the PDB database (https://www.rcsb.org/, accessed on 5 June 2024). Proteins were processed to remove water and ligands using PyMoL software (v2.6). Molecular docking was conducted with the active site centered on the native ligand of the target protein, and results were visualized using PyMoL.

### 2.4. Experimental Animals

Forty-four male C57BL/6J mice (SPF grade), aged 7–8 weeks, were provided by Beijing Speafor Biotechnology Co., Ltd. (Beijing, China) (License No.: SCXK(Jing) 2019-0010). The mice were housed at the Laboratory Animal Center of Dali University (Use License No.: SYXK(Dian) 2024-0001). All experimental and housing conditions strictly adhered to the guidelines for SPF-grade laboratory animals. Animal experiments involved in this study were approved by the Laboratory Animal Ethics Committee of Dali University (Approval No.: 2024-PZ-008).

### 2.5. Establishment of Ap Mouse Model

Mice were anesthetized with isoflurane, and depilatory cream was applied to their dorsal skin [31]. After approximately 2 min, back hair was removed using wet tissues moistened with warm water, creating a 3 cm × 4 cm modeling area. Successful modeling was defined as no residual hair, smooth skin, and no obvious damage. After modeling, mice were randomly divided into four groups: model group (application of the base solution, which is a mixture of water–anhydrous ethanol–1,2-propylene glycol at a ratio of 3:5:2), positive control group (minoxidil, 5% topical minoxidil has been shown to be significantly superior to 2% topical minoxidil for hair regrowth) [32], PA-011 low-dose group (PA-011L), and PA-011 high-dose group (PA-011H). The model group received vehicle solution, the positive control group received 5% minoxidil solution, and the remaining groups received 1% and 4% PA-011 solutions, respectively (the test dose of PA-011 was determined based on the results of preliminary pre-experiments). Treatments were administered twice daily until the end of the experiment on day 21. In this study, PA-011 was administered via external topical application (applied to the depilated area on the back of mice with depilation-induced alopecia model). Both the 1% and 4% concentrations of PA-011 were applied at a volume of 0.2 mL per mouse per application. It should be emphasized that both PA-011 and minoxidil are dissolved in the base solution.

### 2.6. Safety Evaluation

Mice were weighed and their body weights recorded every two days after depilation. On day 21, after sacrificing the mice, major organs including the heart, liver, spleen, lung, and kidney were dissected, weighed, and recorded to calculate organ indices. HE pathological section staining was performed on the major organs of the model group and PA-011 high-dose group.

### 2.7. Observation of Hair Growth Status and Skin Color Scoring

After depilation, hair growth status of mice in each group was observed daily, and skin color was scored every two days using criteria referenced from literature [33]: pink (1 point), pinkish white (2 points), white (3 points), off-white (4 points), gray (5 points), and grayish black (6 points). Photographic records of mice in each group were taken on days 4, 10, 14, 17, and 21 post-depilation.

### 2.8. Evaluation of Hair Growth in Mice

After depilation, the hair-growth-promoting effect of the samples was evaluated every two days with reference to Zhang’s scoring criteria (no growth: 0 points; 0–20% growth: 1 point; 20–40% growth: 2 points; 40–60% growth: 3 points; 60–80% growth: 4 points; 80–90% growth: 5 points; 90–100% growth: 6 points) [34]. On day 21, ten hairs were plucked with forceps from the anterior, middle, and posterior regions of the depilated dorsal area in each group of mice using Kong’s method [35], and hair length was measured with a vernier caliper (mm). Following Gao’s protocol [36], hair within a 3 cm×4 cm depilated area was shaved with a hair clipper and weighed using an analytical balance.

### 2.9. Detection of VEGF and SOD Levels in Skin Tissues

On day 21 of the experiment, after sacrificing the mice, skin tissues from the depilated area of each group were collected, and VEGF and SOD levels in the skin tissues were detected according to the instructions of the ELISA kit.

### 2.10. Blood Routine Analysis

On day 21 of the experiment, mice were fixed with a mouse restrainer, and tail vein blood was collected for routine blood analysis. The measured parameters included the counts of white blood cells (WBCs), red blood cells (RBCs), platelets (PLTs), granulocytes (Grans), monocytes (Mons), lymphocytes (Lyms), and the concentration of hemoglobin (HGB).

### 2.11. Histological and Immunofluorescence Staining

On day 21 of the experiment, skin samples were collected, fixed in 4% paraformaldehyde for 24 h, and paraffin-embedded for HE staining (nuclei stained blue, cytoplasm red). To investigate the number of regenerated hair follicles, we counted the hair follicles in HE-stained pathological sections. According to the experimental design of this study, the sampling areas for hair follicle counting were 3 non-overlapping visual fields selected following the principle of random sampling, with each visual field having an area of 1440 × 1024 mm^2^. Paraffin sections were deparaffinized in xylene for 300 min, treated with absolute ethanol for 15 min, and washed with water. Antigen retrieval was performed using citrate buffer, followed by PBS washing and centrifugation. Sections were incubated with permeabilization solution at room temperature for 20 min and washed three times with PBS (pH 7.4) on a shaker for 5 min each. Slides were prepared according to the TUNEL kit instructions and incubated with Ki67 primary antibody, then visualized under a fluorescence microscope (DAPI: blue nuclei, TUNEL: green, Ki67: red).

### 2.12. Transcriptomic Analysis of Mouse Skin Tissues

Total RNA was extracted from the depilated skin area of mice using Trizol (Invitrogen Life Technologies, Beijing, China), and its concentration, quality, and integrity were measured with a NanoDrop spectrophotometer (Thermo Scientific, Waltham, MA, USA). Using 3 μg of RNA as starting material, mRNA was purified from total RNA using magnetic beads conjugated with poly-T oligonucleotides. mRNA was fragmented by divalent cations at high temperature in an Illumina-specific fragmentation buffer. First-strand cDNA was synthesized using random oligonucleotide primers and SuperScript II, followed by second-strand cDNA synthesis with DNA polymerase I and RNase H, including end processing and enzyme removal. DNA fragments were 3′-adenylated and ligated with Illumina PE adapter oligonucleotides. Library fragments were purified using the AMPure XP system, and cDNA fragments of 400–500 bp were selected. Adapter-ligated DNA fragments were enriched via 15 cycles of PCR using an Illumina PCR Primer Cocktail, and PCR products were purified with the AMPure XP system. Quantification was performed using the Agilent High Sensitivity DNA Assay on an Agilent Bioanalyzer 2100 system. Sequencing was conducted on the DNBSEQ-T7 platform at Shanghai Personal Biotechnology Co., Ltd. (Shanghai, China). After data acquisition, GO enrichment analysis was performed using topGO (*p* < 0.05) to identify biological functions of differentially expressed genes, and KEGG pathway enrichment analysis was carried out using clusterProfiler (3.4.4), with a focus on pathways with *p* < 0.05.

### 2.13. Untargeted Metabolomic Analysis of Mouse Skin Tissues

Fifty milligrams of depilated skin tissue was placed into a 2 mL centrifuge tube, mixed with 200 μL of pre-cooled water and 2 steel beads, and homogenized in a 55 Hz high-throughput tissue grinder for 60 s (repeated once). Eight hundred microliters of methanol–acetonitrile (1:1, *v*/*v*) mixture was added, followed by sonication for 30 min and freezing at −20 °C for 30 min. The mixture was centrifuged at 12,000 rpm for 10 min at 4 °C, and 800 μL of the supernatant was vacuum-concentrated to dryness. The residue was reconstituted with 150 μL of 50% methanol containing 5 ppm 2-chlorophenylalanine, vortexed for 30 s, and centrifuged again at 12,000 rpm for 10 min at 4 °C. The supernatant was filtered through a 0.22 μm membrane into an autosampler vial, and 10–20 μL of each sample filtrate was mixed to prepare quality control (QC) samples for quality assessment.

Chromatographic separation was performed on an ACQUITY UPLC HSS T3 column (Waters, Milford, MA, USA) (100 Å, 1.8 μm, 2.1 mm × 100 mm) at a flow rate of 0.4 mL/min. The column temperature was set at 40 °C, the autosampler temperature at 8 °C, and the injection volume was 2 μL. Data acquisition in positive/negative ion modes was carried out using a Thermo Orbitrap Exploris 120 mass spectrometer (Thermo Fisher Scientific, Waltham, MA, USA) controlled by Xcalibur 4.7 software, operated in data-dependent acquisition (DDA) mode with a heated electrospray ionization (HESI) source (predefined parameters). Both experimental and QC samples were analyzed under the same conditions. The system was equilibrated with 2–4 QC injections before sample analysis, and one QC sample was inserted every 5–10 samples for quality control.

Data were processed using the R package Ropls (v4.5.1) for PCA, PLS-DA, and OPLS-DA dimensionality reduction. Score plots were generated to visualize metabolite differences, and model overfitting was evaluated via permutation tests. Model fitness was assessed using R^2^X, R^2^Y, and Q^2^ values. Differentially expressed metabolites were identified by combining statistical significance (*p* < 0.05), variable importance in the projection (VIP > 1), and fold change (FC). Pheatmap was used for clustering analysis of metabolite abundances, while VennDiagram and UpSetR were employed to visualize differential metabolites. Metabolite correlation analysis was performed using corrplot. Results were further validated via machine learning and ROC curves, and KEGG pathway enrichment analysis was conducted using clusterProfiler (v1.10.1) to identify key metabolic pathways.

### 2.14. Integrated Transcriptomic and Metabolomic Analysis of Mouse Skin Tissues

To investigate the potential mechanisms by which PA-011 improves Ap, integrated analysis of metabolomics and transcriptomics data was performed. First, correlation analysis was conducted between the quantitative results of the two omics. Differentially expressed metabolites and transcripts were then extracted to filter correlation results. Transcripts corresponding to enzymes were retrieved based on metabolite information from the KEGG database, and the differential trends of corresponding differentially expressed metabolites and transcripts were systematically analyzed. Finally, common pathways from the differential enrichment analyses of the two omics were summarized, along with shared annotated pathways of differential metabolites and genes.

### 2.15. Real-Time Fluorescence Quantitative Reverse Transcription PCR (RT-qPCR)

One hundred milligrams of depilated skin tissue was ground and centrifuged to obtain the supernatant, which was treated with chloroform and isopropanol, stored at −20 °C, washed with ethanol, air-dried in a laminar flow hood, and dissolved in nuclease-free water, followed by incubation at 55 °C. The absorbance of 2.5 μL of the sample was measured using a Nanodrop 2000 (Thermo Fisher Scientific, USA) after zeroing, and the RNA concentration was diluted to 200 ng/μL. The PCR machine (Beijing Dongsheng Chuangxin Biotechnology Co., Ltd., Beijing, China) was operated with the following program: 25 °C for 5 min, 42 °C for 30 min, and 85 °C for 5 s. Amplification was performed on a real-time fluorescence quantitative PCR machine using the following protocol: pre-denaturation at 95 °C for 30 s, 40 cycles of denaturation at 95 °C for 15 s, and annealing/extension at 60 °C for 30 s, followed by a melting curve program. The ΔΔCT method was used, where CT values of the target and reference genes were calculated as A and B, respectively. The relative expression was determined as 2^−K^, where K = A − B. mRNA levels were normalized to GAPDH, and primer sequences are listed in Table 1.

### 2.16. 16 s rRNA Detection

Cotton swabs of the depilated skin area from each group of mice were collected on day 21. Samples (0.2–0.5 g) were added to centrifuge tubes containing extraction lysis buffer and homogenized using a multi-sample tissue grinder (Shanghai Jingxin, China, Tissuelyser-48) at 60 Hz. Following pretreatment, nucleic acids were extracted using the OMEGA Soil DNA Kit (M5635-02). The extracted DNA was assessed for molecular size by 0.8% agarose gel electrophoresis and quantified by Nanodrop. PCR amplification targeted the bacterial 16S rRNA V3–V4 region using primers 338F (5′-barcode+ACTCCTACGGGAGGCAGCA-3′) and 806R (5′-GGACTACHVGGGTWTCTAAT-3′). PCR products were quantified using the Quant-iT PicoGreen dsDNA Assay Kit on a Microplate Reader (BioTek, Winooski, VT, USA, FLx800) before pooling. Libraries were prepared using the Illumina TruSeq Nano DNA LT Library Prep Kit, and qualified libraries were quantified with the Quant-iT PicoGreen dsDNA Assay Kit on a Promega QuantiFluor. Sequencing was performed on an Illumina NovaSeq 6000 SP (500 cycles) using 2 × 250 bp paired-end reads. Partial analyses (Alpha and Beta diversity, taxonomic composition, intergroup differences, co-occurrence network construction, and microbial metabolic function prediction) were completed via the Personal Gene Cloud platform.

### 2.17. Statistical Analysis

Data were analyzed using SPSS 19.0 statistical software and presented as mean ± standard error of the mean (SEM). Statistical analyses and visualization were performed using GraphPad Prism 10. Independent samples *t*-tests were used for comparisons between two groups, while one-way analysis of variance (one-way ANOVA) was applied for intergroup difference analyses involving more than two groups. Statistical significance was defined as *p* < 0.05. Results compared with the model group are denoted as * *p* < 0.05, ** *p* < 0.01, *** *p* < 0.001, and **** *p* < 0.0001. Partial analyses for transcriptomics, metabolomics, and microbiomics were completed via the Personal Gene Cloud platform (https://www.genescloud.cn/analysis/diversityAnalysis, accessed on 23 May 2024).

## 3. Results

### 3.1. Network Pharmacology Results

#### 3.1.1. Identification of Core Targets

LC-MS/MS analysis revealed that PA-011 contained 344 small-molecule compounds and 16,788 peptide sequences, including 2055 database peptides and 14,460 de novo only peptides. Using the set peptide screening criteria, 33 peptide sequences were selected from the top 100 database peptides based on −10 lg *p* values, and 102 peptide sequences were selected from 287 de novo only peptides with a confidence > 95%. After merging and removing duplicates, a total of 129 peptide sequences were obtained.

A total of 474 drug targets were predicted from 157 small-molecule compounds and 129 peptide sequences in PA-011. Integration of disease database information yielded 5002 disease target entries. Venn diagram analysis (Figure 1A) identified 147 intersecting targets from the integrated data.

#### 3.1.2. PPI Analysis of Core Targets

PPI network analysis of the obtained intersecting targets yielded a PPI network diagram (Figure 1B) with 143 nodes and 1352 edges. Using the default screening criteria of the software plugin, a protein–protein interaction network with 28 core protein nodes and 285 edges was filtered (Figure 1C). Darker node colors indicate higher degree values, while darker and denser line colors signify stronger protein connectivity. Core proteins with positive effects on Ap treatment, such as STAT3, IL-1β, Bcl-2, Akt, and ESR1, exhibited strong connectivity.

#### 3.1.3. Enrichment Analysis

GO enrichment analysis (Figure 1D) and KEGG enrichment analysis (Figure 1E) of the intersecting targets revealed that biological process (BP) terms from GO analysis indicated components in PA-011 positively regulated therapeutic objectives such as negative regulation of gene expression in Ap, positive regulation of cell migration, positive regulation of cell proliferation, positive regulation of apoptotic processes, and positive regulation of angiogenesis. cellular component (CC) and molecular function (MF) analyses showed that PA-011 primarily acted on intercellular signaling, participated in apoptotic processes, and exerted protective effects against Ap through amino acid and peptide activity.

KEGG enrichment analysis of pathway information further showed that PA-011 regulated signaling pathways influencing apoptosis, including the MAPK, PI3K-Akt, and Foxo pathways, and exerted multiple therapeutic effects on Ap, such as anti-inflammation, anti-apoptosis, regulation of cellular metabolism, and control of steroid hormone activity.

#### 3.1.4. Drug-Target Pathway Correlation Network Analysis

Network visualization (Figure 2) was performed using KEGG target information, intersecting protein data, and peptide data. As shown in the correlation map, small-molecule and peptide components in the drug exhibited high connectivity with core targets such as STAT3, ACE, STAT6, and C5AR1. Enriched pathways included lipid and atherosclerosis (hsa05417), endocrine resistance (hsa01522), steroid hormone biosynthesis (hsa00140), and the AGE-RAGE signaling pathway in diabetic complications (hsa04933). The dense connections among these five elements indicated a high degree of correlation between PA-011 and AGA treatment targets.

#### 3.1.5. Analysis of Molecular Docking Results

Binding energy from molecular docking indicated that lower binding energy between ligand and receptor signified more stable binding and stronger binding activity. In the protein–protein interaction (PPI) analysis, nine target proteins with relatively high degree values and strong relevance to alopecia were selected for docking, namely AKT1 (3cqu), ESR1 (1YIM), IL1B (3POK), BCL2 (2O22), TGFB1 (6P7J), CASP3 (1RHJ), PPARG (2GTK), AR (2AM9), MPO (5MFA), along with 16 small molecules (Table 2). Additionally, nine target proteins and the top ten peptide sequences by degree value (Table 3) were docked under the same conditions. Results showed that 100% of small molecules had binding energies < −5 kcal/mol with their targets, and 79.86% had binding energies < −7.0 kcal/mol. For peptides, 100% of sequences had binding energies < −5 kcal/mol, and 83.33% had binding energies < −7.0 kcal/mol. Docking results were visualized as heatmaps (Figure 3A,H), indicating strong binding activity between PA-011 active components and Ap targets. Selected small molecules and peptides with binding energies < −9 kcal/mol were visualized using PyMoL software (Figure 3B–G,I–N).

### 3.2. Safety Evaluation of PA-011

Body weights of mice in each group were measured every two days starting from the day of depilation. Results showed that body weights exhibited an upward trend over time, with a slight decrease only on the day of modeling. Notably, significant differences in body weight were observed between the PA-011 intervention and model groups from day 5 to day 17 (Figure 4B, *p* < 0.05, *p* < 0.01, *p* < 0.001, or *p* < 0.0001). Organ index analysis revealed no significant differences between the PA-011 intervention and model groups (Figure 4C, *p* > 0.05).

HE staining of various organs (Figure 4A) showed the following: heart: intact myocardial structure, clear muscle fibers, and normal cell arrangement; liver: intact capsule, well-defined hepatic lobule structure, abundant hepatocyte cytoplasm, and large round nuclei, consistent with normal morphological characteristics; spleen: intact capsule and trabeculae, clear red/white pulp boundary, and normal splenic corpuscle morphology, volume, and quantity; lung: clear bronchial, alveolar septum, and epithelial cell structures, with no thickening of the alveolar walls; kidney: intact capsule, normal glomerular and tubular structures, and no obvious lesions compared with the model group. These results indicate that topical application of PA-011 causes no apparent organ toxicity, suggesting it may be a safe candidate for Ap treatment.

### 3.3. PA-011 Significantly Promotes Hair Regeneration in Depilated Mouse Skin

Observations and scoring of hair regrowth in depilated mice showed that PA-011 significantly enhanced hair regeneration (Figure 5A). The hair growth cycle changes with skin color, and the hair growth cycle can be reflected by observing skin color. The skin color scoring results (Figure 5B) showed that from day 5 to day 17 after depilation, there were significant differences between the minoxidil group, PA-011 treatment groups and the model group (*p* < 0.05, *p* < 0.01, *p* < 0.001 or *p* < 0.0001). Hair coverage scores (Figure 5C) demonstrated extremely significant differences in the minoxidil and PA-011 groups compared to the model group from day 9 onward (*p* < 0.001 or *p* < 0.0001). At day 21, hair length and weight (Figure 5D,E) in the minoxidil and PA-011 groups also showed significant differences compared to the model group (*p* < 0.01, *p* < 0.001, or *p* < 0.0001). Collectively, these results indicate that PA-011 treatment effectively alleviates HF damage caused by depilation, promotes rapid entry of HFs into the anagen phase, and thereby accelerates hair growth in mice.

### 3.4. PA-011 Significantly Promotes HF Cell Proliferation

Mice were euthanized 21 days after depilation, and skin tissues from the dorsal depilated area were collected for HE staining analysis. Morphological observations (Figure 6A) showed significant differences in HF number and morphology between the experimental groups and the model group. Both the minoxidil and PA-011 intervention groups significantly improved HF structure and enhanced the functional activity of dermal papilla cells. Quantitative analysis revealed that compared with the model group, the minoxidil and PA-011 groups exhibited significantly increased numbers of vellus follicles (Figure 6B), terminal follicles (Figure 6C), and total HFs (Figure 6D) (*p* < 0.0001), as well as a significantly higher terminal-to-vellus follicle ratio (Figure 6E, *p* < 0.05).

Dynamic changes in HF cells were further detected via TUNEL/Ki67 immunofluorescence double staining (Figure 6F). Results showed that the PA-011 intervention group had a significantly lower proportion of apoptotic cells (Figure 6G, *p* < 0.05) and a significantly higher proportion of proliferating cells (Figure 6H, *p* < 0.05). These findings indicate that PA-011 may effectively increase the number of active HFs and prolong the hair growth cycle through a dual regulatory mechanism: inhibiting follicular cell apoptosis and promoting cell proliferation.

### 3.5. PA-011 Increases VEGF and SOD Expression Levels in Skin Tissues

Compared with the model group, the minoxidil and PA-011 intervention groups exhibited significantly higher protein expression levels of VEGF in skin tissues (Figure 6I). The minoxidil group showed a statistically significant difference (*p* < 0.05), while the PA-011 group exhibited an extremely significant difference (*p* < 0.01). Although the expression level of SOD did not reach statistical significance (Figure 6J, *p* > 0.05), both groups showed an upward trend. These results suggest that PA-011 may promote hair growth in Ap mice by enhancing VEGF expression in skin tissues.

### 3.6. Effects of PA-011 on Peripheral Blood Parameters in Mice

After 21 days of PA-011 administration, the model group showed abnormally elevated WBC counts in peripheral blood, exceeding normal values. Compared with the model group, the PA-011 intervention group had significantly reduced numbers of WBCs, Grans, Mons, and Lyms (Figure 7A, *p* < 0.05, *p* < 0.01, or *p* < 0.001). However, there were no significant differences in RBC counts (Figure 7B), PLT counts (Figure 7C), or HGB concentration (Figure 7D) (*p* > 0.05). These findings indicate that depilation induces skin micro-inflammation, leading to elevated WBC counts, which can be effectively mitigated by PA-011.

### 3.7. Transcriptomic Results of Skin Tissues

#### 3.7.1. Sequencing Data Quality Control and Basic Analysis

To elucidate the molecular mechanisms underlying PA-011-induced hair growth, transcriptome sequencing was performed on skin tissues (*n* = 4). Quality control analysis showed uniform coverage of gene regions by sequencing data (Figure 8A), meeting the requirements for deep sequencing. FPKM density distribution indicated that moderately and highly expressed genes were dominant (Figure 8B), consistent with the characteristics of mammalian tissue transcriptomes. Principal component analysis (PCA) revealed distinct clustering between the model and PA-011 groups (Figure 8C). Pearson correlation coefficients confirmed significant intergroup differences (r < 0.8) and good intragroup reproducibility (r > 0.8) (Figure 8D), satisfying the conditions for subsequent differential analysis.

#### 3.7.2. Screening of Differentially Expressed Genes and Network Construction

DESeq2 analysis identified 1346 differentially expressed genes (|log_2_FC| > 1, *p* < 0.05), including 918 upregulated and 428 downregulated genes (volcano plot in Figure 8E). Hierarchical clustering showed significant expression separation of differentially expressed genes between the two groups (heatmap in Figure 8F). Genomic circle plots were used to visualize the expression of all differentially expressed genes (Figure 8G). Through protein–protein interaction (PPI) network analysis with a set PPI score threshold > 0.95, we identified key regulatory nodes: compared with the PA-011H group, the follicle inhibitory factor DKK2 was significantly downregulated in the model group, while pro-inflammatory factors such as IL1α and IL1R2 were notably upregulated (Figure 8H). These results suggest that PA-011 may promote hair follicle (HF) regeneration by inhibiting the inflammatory microenvironment.

#### 3.7.3. Functional Enrichment and Pathway Analysis

GO enrichment analysis showed that differentially expressed genes were significantly involved in biological processes such as epidermal development (*p* = 5.86 × 10^−27^) and keratinocyte differentiation (*p* = 2.65 × 10^−30^) (Figure 8I). KEGG pathway analysis revealed that PA-011 exerted its effects through key pathways, including the FoxO signaling pathway (*p* = 0.014), estrogen signaling (*p* = 0.001), and arachidonic acid metabolism (*p* = 0.007) (Figure 8J). Notably, significant enrichment of α-linolenic acid (*p* = 0.0003) and linoleic acid metabolism (*p* = 0.02) suggested that regulation of the lipid microenvironment may be an important mechanism of action for PA-011.

### 3.8. Metabolomic Analysis of Skin Tissues

#### 3.8.1. Metabolite Composition Analysis of Skin Tissue

Based on LC-MS metabolomics detection, six major classes of metabolites were identified under positive (POS) and negative (NEG) modes (Figure 9A). Lipids and lipid-like molecules accounted for the highest proportion (33.04%), followed by organic acids and their derivatives (27.8%) and amino acids and peptides (10.7%), suggesting that lipid metabolism reprogramming may play a critical role in the mechanism of PA-011. Multidimensional statistical analysis was used to evaluate metabolic differences between groups: PCA analysis showed distinct separation trends between the model and PA-011H groups under both POS/NEG modes (Figure 9B). Supervised learning models PLS-DA (Figure 9C) and OPLS-DA (Figure 9D) further validated the metabolic profile differences between groups (Q^2^ > 0.5), confirming that the data reliability met the criteria for differential metabolite screening.

#### 3.8.2. Screening of Differential Metabolites and Functional Association

Hierarchical clustering heatmaps (Figure 9E) visually demonstrated significant separation of metabolite expression profiles between the two groups, with a total of 327 differential metabolites identified (VIP > 1, *p* < 0.05). Notably, the differential lipid metabolites corresponded to the transcriptional changes in α-linolenic acid and arachidonic acid metabolic pathways, suggesting that PA-011 may promote HF regeneration through a dual network of “gene expression-metabolic regulation.”

#### 3.8.3. Correlation of Differential Metabolites

Differential metabolites were screened and subjected to enrichment analysis. A random forest algorithm was used to identify key differential metabolites, revealing seven major metabolites under POS mode: 6,7-dimethoxycoumarin, 7-methylimidazo [1,2-a] pyrimidin-5 (1H)-one, 4-(dimethylamino) phenylalanine, etc. (Figure 10A). Correlation analysis showed that 6,7-dimethoxycoumarin was positively correlated with the other six metabolites (Figure 10B). Under NEG mode, core differential metabolites included seven substances such as salicyluric acid, acetylleucine, and 3-hydroxy-5-methoxy-3-methyl-5-oxopentanoic acid (Figure 10C), among which salicyluric acid was positively correlated with the remaining six metabolites (Figure 10D).

#### 3.8.4. Functional Enrichment Analysis of Differential Metabolites

KEGG enrichment analysis was performed on differential metabolites, and the top 20 pathways were visualized: under positive ion mode, model_vs_PA-011H differential metabolites were primarily enriched in calcium signaling pathway, oxidative phosphorylation, nicotinate and nicotinamide metabolism, NOD-like receptor signaling pathway, cGMP-PKG signaling pathway, and necroptosis (Figure 10E,F). Under negative ion mode, enrichment was mainly observed in AMPK signaling pathway, mTOR signaling pathway, PI3K-Akt signaling pathway, and FoxO signaling pathway (Figure 10G,H).

Integrated metabolomics–transcriptomics analysis identified a key converging pathway: FoxO signaling (*p* = 0.026 in metabolomics; *p* = 0.014 in transcriptomics) was significantly enriched in both omics. This finding suggests that PA-011 may regulate HF energy-lipid metabolism balance through the “FoxO-α-linolenic acid metabolic axis,” providing molecular evidence for its multi-target mechanism of action.

### 3.9. Multi-Omics Integration Reveals Core Action Pathways of PA-011

#### 3.9.1. Construction of Cross-Omics Correlation Model

Orthogonal partial least squares (OPLS) was used to integrate transcriptomic and metabolomic data, with a model interpretability of R^2^ > 0.7 (Figure 11E), indicating a significant co-regulatory relationship between the two omics datasets. The top 20 key associated molecules (genes/metabolites) were screened by loadings analysis, among which 2′,4′-Dihydroxy-2-biphenylcarboxylic_acid (Loading = 0.029) and Krtap4-9 (Loading = 0.011) showed the strongest co-variation trend (Figure 11A), suggesting a functional coupling between lipid metabolites and HF development genes.

#### 3.9.2. Analysis of Gene-Metabolite Interaction Networks

Based on Pearson correlation coefficients (|r| > 0.8), the top 100 correlation pairs were constructed (Figure 11B). Core metabolite lapachol was significantly positively correlated with antioxidant genes such as FoxO1 and SOD2, while salicyluric acid was significantly negatively correlated with IL1RN. This finding forms a mechanistic loop with previous results of IL1 family downregulation in transcriptomics and antioxidant metabolite upregulation in metabolomics.

#### 3.9.3. Validation of Pathway Cascade Regulation

Through KEGG enzyme-gene mapping analysis, the FoxO signaling pathway was identified as a key convergence node (Figure 11C). At the metabolic level, differential metabolites were enriched in AMPK-FoxO and mTOR-FoxO regulatory modules. At the transcriptional level, genes such as FoxO1 and SIRT1 were significantly upregulated. Functionally, the FoxO pathway enhanced ROS scavenging capacity by activating SOD2 and alleviated inflammatory damage by inhibiting IL1R2, forming a dual “antioxidant-anti-inflammatory” protective mechanism.

Integrating multi-omics evidence, a functional model of PA-011 was proposed (Figure 11D): by activating the AMPK/FoxO signaling axis, it synchronously regulates α-linolenic acid metabolic reprogramming and the antioxidant defense system of HF stem cells, thereby prolonging the HF anagen phase and improving microenvironmental homeostasis.

### 3.10. RT-qPCR Validation of Core Genes

To validate whether PA-011 exerts its hair-promoting effect through the FoxO pathway, RT-qPCR was performed to analyze genes associated with the FoxO pathway. Results showed that compared with the model group, PA-011 intervention significantly increased the expression levels of Foxo3, Akt1, Pik3r3, and Pik3r1 (Figure 12A–D) (*p* < 0.05, *p* < 0.01, or *p* < 0.0001), while the expression levels of Fasl and Ccnd1 (Figure 12E,F) were significantly decreased (*p* < 0.05 or *p* < 0.01). These findings confirm that PA-011 promotes hair growth by regulating the FOXO/PI3K/AKT signaling pathway.

### 3.11. Skin Microbiome Analysis

#### 3.11.1. Alpha Diversity Analysis of Skin Microbiome

Alpha diversity analysis of sequencing data from model and PA-011H group samples (*n* = 4) showed significant differences in Simpson’s index and Pielou’s evenness index between model_vs_PA-011H groups (Figure 13A, *p* < 0.05). By plotting the rarefaction curve to analyze the trend of Alpha diversity with sampling depth (Figure 13B), microbial abundance in the model group was significantly higher than that in the PA-011H group, indicating that PA-011 treatment significantly altered skin microbial abundance and balance.

#### 3.11.2. Beta Diversity Analysis of Skin Microbiome

Beta diversity analysis was used to explore the dissimilarity in species composition between the two groups. Principal coordinates analysis (PCoA) showed that samples from the model and PA-011H groups were distantly separated, while intragroup samples were closely clustered (Figure 13C). Non-metric multidimensional scaling (NMDS) had a stress value of 0.0000984, indicating reliable results (Figure 13D). A Venn diagram analysis revealed 221 shared microbial species between the two groups, with the model group containing significantly more microbial species than the PA-011H group (Figure 13E), further confirming that PA-011 treatment significantly altered skin microbiome balance and species composition.

#### 3.11.3. Skin Microbiome Species Analysis

Further analysis of abundance changes in skin microbiota between the model and PA-011H groups was performed for the top 10 microbial taxa at the phylum, class, order, family, genus, and species levels. Results showed that at the phylum level (Figure 14A), PA-011H intervention significantly increased the abundance of *Verrucomicrobiota* (*p* < 0.001) and decreased *Patescibacteria* (*p* < 0.05). At the class level (Figure 14B), PA-011H significantly enhanced *Kiritimatiella abundance* (*p* < 0.01) and reduced *Saccharimonadia* (*p* < 0.05). At the order level (Figure 14C), PA-011H significantly increased *Pseudomonadales abundance* (*p* < 0.01). At the family level (Figure 14D), PA-011H significantly elevated *Moraxellaceae* (*p* < 0.01) and *Aerococcaceae* (*p* < 0.05). At the genus level (Figure 14E), PA-011H significantly increased *Aerococcus abundance* (*p* < 0.05). At the species level (Figure 14F), PA-011H significantly enhanced *Acinetobacter schindleri* (*p* < 0.0001) and *Acinetobacter variabilis* (*p* < 0.05), indicating that PA-011H regulates skin microbial abundance.

Random forest analysis was used to screen marker species, and the top 10 most important species were visualized: at the phylum level (Figure 14G), *Verrucomicrobiota* and *Firmicutes_B* were key species; at the class level (Figure 14H), *Actinomycetia* and *Kiritimatiellae* were major marker species; at the order level (Figure 14I), *Sphingobacteriales* and *Erysipelotrichales* ranked highly in importance; at the family level (Figure 14J), *Enterococcaceae* and *Moraxellaceae* were core marker families; at the genus level (Figure 14K), *Formimonas* and *Ligilactobacillus* were key genera; at the species level (Figure 14L), *Bogoriella caseilytica* and *Atopostipes suicloacalis* were major marker species.

#### 3.11.4. Biomarkers of Differential Skin Microbiome Between Groups

Linear discriminant analysis effect size (LEfSe) was used to screen for microorganisms with significant differences (LDA threshold = 2) between the model and PA-011H groups (Figure 14M,N). The model group had 17 dominant species, with the top five LDA values observed in g_Kineothrix, f_Carnobacteriaceae, c_Desufovlbrionia, f_Desulfovibrionaceae, and p_Desulfobacterota_I. The PA-011H group had 15 dominant species, with the top five LDA values in g_Acinetobacter, o_Pseudomonadales, f_Moraxellaceae, g_Aerococcus, and f_Aerococcaceae.

#### 3.11.5. Predictive Analysis of Metabolic Pathways in Differential Bacterial Species

Functional potential prediction of differential microbiome (Figure 15A) revealed that the metabolic functions of skin microbiota were primarily concentrated in amino acid biosynthesis, carbohydrate biosynthesis, cell structure biosynthesis, cofactor/prosthetic group/electron carrier and vitamin biosynthesis, fatty acid and lipid biosynthesis, nucleoside and nucleotide biosynthesis, glycolysis, and L-aspartate and L-asparagine biosynthesis superpathways. In the model_vs_PA-011H group, significantly different metabolic pathways included PWY0-41 (allantoin degradation IV), PWY-7398 (coumarin biosynthesis), and CODH-PWY (reductive acetyl-CoA pathway) (Figure 15B). The PWY0-41 pathway was primarily contributed by Aerococcus, Acinetobacter, Saccharopolyspora_C, and Atopostipes (Figure 15C); the PWY-7398 pathway mainly involved Cupriavidus, Promicromonospora, Pseudomonas_E, and Pseudonocardia (Figure 15D); and the core species of the CODH-PWY pathway included unclassified Oscillospiraceae, Ventrimonas, Desulfovibrio_R, and Enterenecus (Figure 15E). These results indicate that PA-011 may improve skin health and promote hair growth by regulating the skin microbiome and intervening in biological metabolic processes such as amino acid, carbohydrate, and cell structure metabolism.

## 4. Discussion

Hair loss, also known as alopecia, is a pathological condition. Its main characteristic is abnormal or excessive hair loss. Its pathogenesis involves multiple factors, including androgen abnormalities, genetic susceptibility, neuroendocrine disorders, drug effects, physical-chemical stimulation, and immune regulation imbalances [37,38], although the specific molecular mechanisms remain to be fully elucidated. Current clinical intervention strategies mainly include surgical transplantation, drug regulation, phototherapy, and cosmetic modification. However, these methods have significant limitations: surgical therapy struggles to achieve ideal hair density reconstruction and cannot reverse the pathological process [39]; first-line clinical drugs (finasteride, minoxidil) can improve symptoms short-term but are associated with severe adverse effects such as sexual dysfunction and high recurrence risk [40,41].

Given the multi-component, multi-target, and low-side-effect properties of traditional Chinese medicine (TCM), exploring the application of low-toxicity functional natural products in Ap prevention and treatment has become a research hotspot. In recent years, TCM has demonstrated promising effects in Ap treatment, with its safety widely recognized [42]. With the rise of network pharmacology and the continuous improvement of omics technologies, the research pathway of exploring the mechanisms of TCM’s complex systems through multi-omics integration has become increasingly clear, providing more scientific and reliable technical support for TCM modernization.

Existing studies have revealed the hair-growth-promoting mechanisms of various natural extracts. For example, Rosa rugosa extract exhibits hair growth effects in DHT-induced Ap mouse models [12]; Pilose antler extract improves hair growth in androgenetic Ap mice by promoting the initial anagen phase [43]; the hair-growth-promoting effect of soluble sturgeon fish oil correlates with the gut microbiome [44]; Platycladus orientalis leaf extract enhances hair growth by activating non-receptor tyrosine kinase ACK1 [45]; Ganoderma lucidum extract attenuates corticotropin-releasing hormone-induced senescence of human HF cells [46]; and Lithospermum erythrorhizon extract deoxyshikonin promotes hair growth by targeting the Wnt/β-catenin signaling pathway [5]. These findings highlight the substantial research value and application potential of traditional Chinese medicine (TCM) in Ap treatment.

Modern pharmacological studies have shown that the PA extract possesses multiple biological activities, including antioxidation [24], anti-inflammation [25], antibacterial effects [26], promotion of tissue repair [27], anti-tumor activity [47], enhanced myocardial protection [47], promotion of wound healing [48], and angiogenesis [49]. These properties are highly congruent with the core objectives of Ap treatment. Based on the remarkable tissue repair-promoting and anti-inflammatory activities of PA extract, this study hypothesized that its derivative PA-011 might promote hair growth in Ap mice.

Hair growth can be regulated through modulation of the hair cycle, such as prolonging the anagen phase and promoting the transition from telogen to anagen [50]. HFs in C57BL/6 mice contain melanocytes, with melanin synthesis synchronized to the hair growth cycle, allowing characterization of the hair cycle by monitoring skin color changes (from pink to black) [51]. The dorsal skin of telogen-phase mice appears pink, subsequently transitioning to gray and black [52]. Experiments showed that skin color scores and hair coverage in the PA-011 treatment group were significantly higher than those in the model group, confirming that PA-011 promotes the telogen-to-anagen transition. HE pathological staining revealed that the number of HFs and the ratio of terminal to vellus follicles in the PA-011 group were significantly higher than in the model group, suggesting that PA-011 may prolong the interval between anagen and telogen. Immunofluorescence results indicated that the TUNEL-positive expression rate in the PA-011 intervention group was significantly lower, while the Ki67-positive expression rate was significantly higher, than in the model group, indicating that PA-011 reduces follicular cell apoptosis and promotes cell proliferation.

Cutaneous blood flow influences the HF cycle (telogen, anagen, catagen) [12], and growth factors such as VEGF regulate the hair growth cycle [53]; their imbalance can lead to Ap. VEGF is a key regulator of cutaneous angiogenesis [54], with active angiogenesis in anagen follicles and regression in catagen follicles [55], as anagen follicles are highly dependent on VEGF for vascular formation [56]. This study found that VEGF expression increased in Ap mice after PA-011 treatment, suggesting that topical PA-011 may promote hair growth by enhancing cutaneous microvascular formation.

Based on the above experimental evidence, this study used LC-MS combined with peptidomics to analyze the composition of PA-011 and integrated network pharmacology models to predict its active components and potential mechanisms for promoting hair growth. Results showed that PA-011 can act on core proteins related to Ap treatment, such as Bcl-2, Akt, and ESR1. GO and KEGG enrichment analyses indicated that it primarily regulates the MAPK signaling pathway, PI3K-Akt signaling pathway, and FoxO apoptosis regulatory pathway, exerting anti-Ap effects through anti-inflammation, anti-apoptosis, and regulation of cell metabolism. Through molecular docking binding energy screening, 14 key active compounds were identified, including (R)-Methysticin, (S)-N-Methylcoclaurine, peptide70, and peptide76.

Integrated transcriptomics and metabolomics analysis showed that the hair-promoting effect of PA-011 was significantly associated with FoxO pathway regulation, consistent with network pharmacology predictions. FOXO transcription factors participate in hair growth regulation by modulating multi-dimensional biological functions such as inflammation, oxidative stress, apoptosis, proliferation, and metabolism [57,58]. As downstream targets of the IRS1/PI3K/AKT signaling pathway, they regulate cell proliferation, apoptosis, and insulin sensitivity through the IRS1/PI3K/FOXO axis [59]. FOXO3 specifically regulates follicular embryonic endothelial development and promotes HF density and hair growth under chronic stress via PI3K/Akt pathway activation [1,42,60]. This study is the first to demonstrate the efficacy of PA extract PA-011 in a depilatory cream-induced Ap model, with mechanisms including (1) promotion of hair growth gene expression, such as upregulation of follicle regeneration-related genes *Foxo3*, *Akt1*, *Pik3r3*, and *Pik3r1*; (2) inhibition of Ap-related genes, such as downregulation of follicle degeneration-promoting genes *Fasl* and *Ccnd1*.

These findings indicate that topical application of PA-011 significantly accelerates hair regeneration and increases hair density in depilated mice, providing multi-omics evidence to support its clinical application.

One of the key triggers of Ap is scalp microbiome dysregulation induced by HF infection and inflammation, which contributes to hair loss by interfering with follicular immunological properties, circulatory function, and regenerative capacity [61,62]. Focusing on the skin microbiome, this study found significant differences in scalp microbial abundance between the model and PA-011 intervention groups: PA-011 intervention significantly reduced the abundance of Saccharimonadia, a taxon closely associated with inflammatory Ap, suggesting it may alleviate perifollicular inflammation by inhibiting pro-inflammatory microbes. Additionally, protective microbiomes such as Verrucomicrobiota, Pseudomonadales, Moraxellaceae, Aerococcus, and Acinetobacter were enriched [63]. Their metabolites can regulate keratinocyte proliferation, indirectly promoting the telogen-to-anagen transition of HFs, inhibiting excessive anaerobic bacterial proliferation, improving the hypoxic microenvironment of HFs, and exerting anti-inflammatory effects by suppressing pathogen colonization, thereby stabilizing skin microbiota ecology, maintaining HF health, and promoting hair regeneration [63].

Notably, the Kineothrix genus, which was specifically overexpressed in the model group, was undetectable or significantly downregulated after PA-011 intervention. This genus is positively correlated with the expression of skin barrier damage markers (e.g., Filaggrin gene), and its abundance changes may serve as microbial biomarkers for early Ap diagnosis. Combining with the reported “skin inflammation → microbiome dysregulation → follicular regeneration inhibition” vicious cycle [64], PA-011 may synergistically promote hair growth through triple mechanisms of anti-inflammation, improvement of follicular microenvironment, and barrier repair by reshaping scalp microbiome structure (e.g., increasing Verrucomicrobiota/decreasing Firmicutes ratio).

While this study has revealed the potential mechanisms of PA-011 in promoting hair growth through multi-omics techniques, several limitations remain. For example, it lacks diversified validation across models, as the current research only includes intuitive efficacy evaluations on animal skin surfaces. Future experiments will incorporate human HF in vitro culture systems to systematically assess the universality and dose–effect relationships of PA-011. Additionally, CRISPR-Cas9 technology will be used to generate knockout mice for key genes such as FoxO3 and Akt1, combined with proteomics to dynamically track regulatory nodes in the HF cycle and map a precise “compound-target-pathway” regulatory network.

Notably, no obvious blood or visceral toxicity was observed after PA-011 intervention. Subsequent studies will conduct 90-day repeated-dose toxicity tests and reproductive toxicity assessments to confirm the clinical safety of PA-011, enhance its clinical translation potential, and provide a safe and efficient new TCM-based solution for Ap treatment. This work also offers methodological references for the development of insect-derived drugs.

## 5. Conclusions

PA extract PA-011 effectively promotes hair growth in depilatory cream-induced Ap model mice. Its mechanisms of action involve regulation of the FOXO/PI3K/AKT signaling pathway, improvement of skin microbiome balance, enhancement of HF cell proliferation, modulation of the hair cycle, and upregulation of VEGF expression. Additionally, based on changes in white blood cell counts, an anti-inflammatory effect may represent another potential mechanism of PA-011 in Ap treatment. Topical application of PA-011 showed no hematological or organ toxicity, indicating good safety.

In summary, PA-011 exhibits multi-dimensional pharmacological mechanisms, combining efficacy and safety, and possesses development potential as a novel drug or functional ingredient for Ap treatment. Future studies are needed to further validate its effects on clinical types of Ap such as androgenetic Ap, and to clarify its core active components and long-term toxicity profile, thereby facilitating translational application.

## Figures and Tables

**Figure 1 cimb-47-00619-f001:**
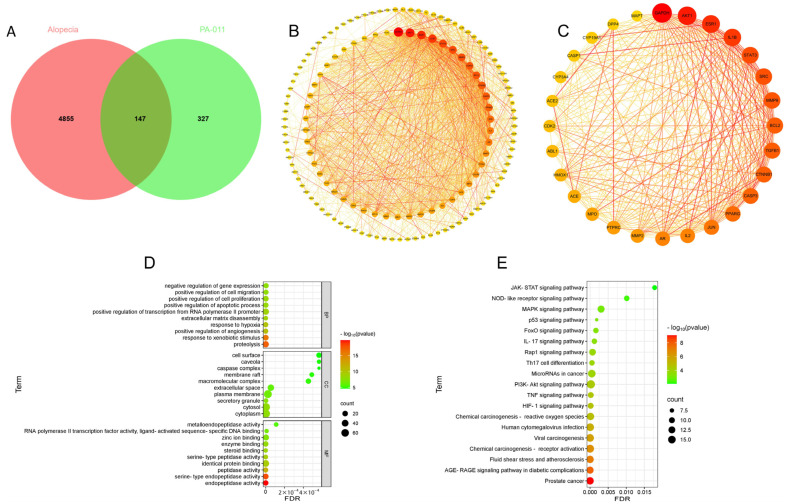
Network pharmacology predicts the hair-regenerative mechanism of PA-011. (**A**) Venn diagram of intersecting targets; (**B**) PPI network diagram; (**C**) core targets; (**D**) bubble plot of GO enrichment analysis; (**E**) bubble plot of KEGG enrichment analysis.

**Figure 2 cimb-47-00619-f002:**
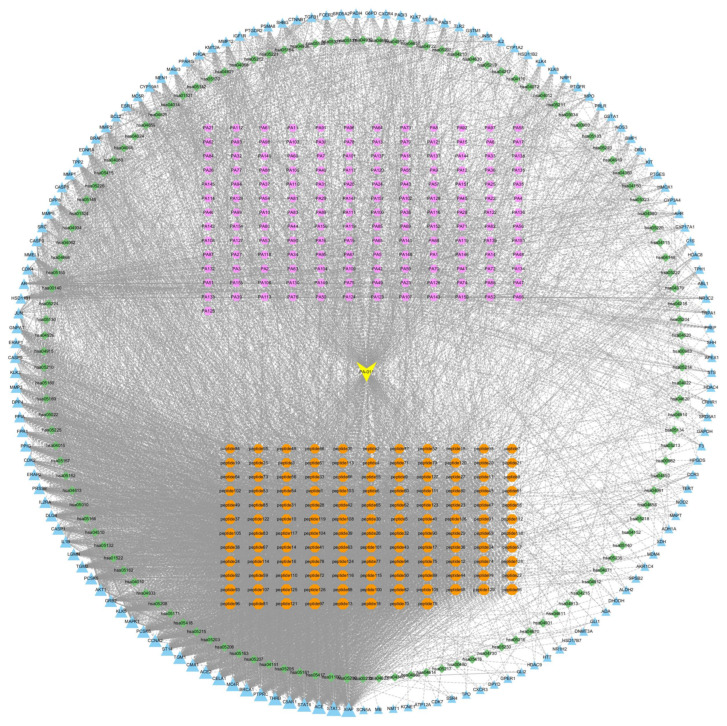
Interaction network of PA-011-disease-pathways. Yellow represents PA-011 samples, orange denotes disease targets, blue indicates small-molecule compounds, purple signifies peptides, and green corresponds to KEGG pathways.

**Figure 3 cimb-47-00619-f003:**
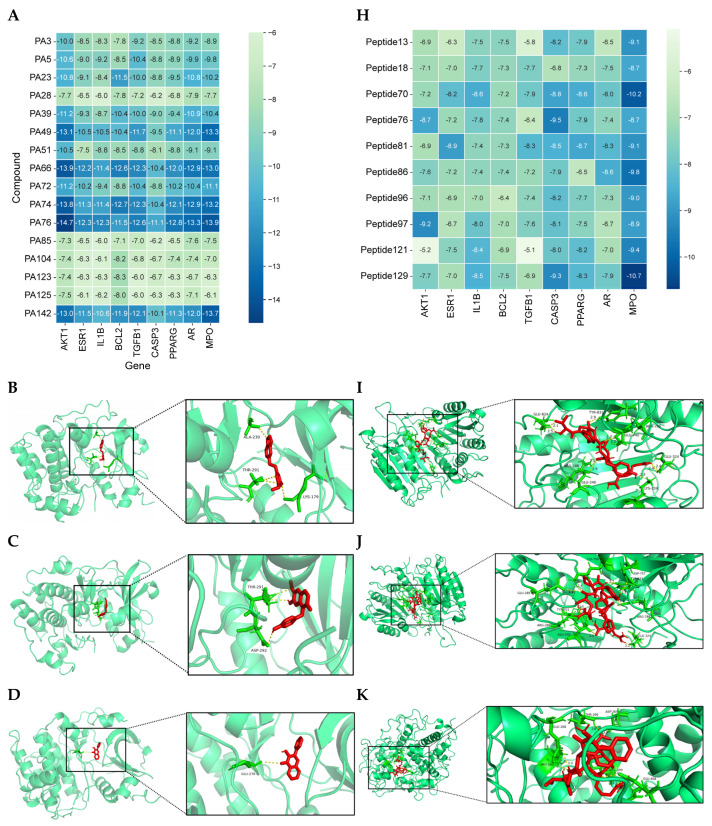
Molecular docking heatmaps of PA-011. (**A**) Heatmap of small-molecule compound docking; (**B**) AKTI docked with (R)-Methysticin; (**C**) AKTI docked with (S)-N-Methylcoclaurine; (**D**) AKTI docked with 3-Hydroxyflavone; (**E**) AR docked with 3-Hydroxyflavone; (**F**) AKT1 docked with 5,7-Dihydroxyflavone; (**G**) ESR1 docked with 5,7-Dihydroxyflavone; (**H**) Heatmap of polypeptide molecular docking scores; (**I**) CASP3 docked with peptide76; (**J**) CAPS3 docked with peptide129; (**K**) MPO docked with peptide70; (**L**) MPO docked with peptide86; (**M**) MPO docked with peptide121; (**N**) MPO docked with peptide129.

**Figure 4 cimb-47-00619-f004:**
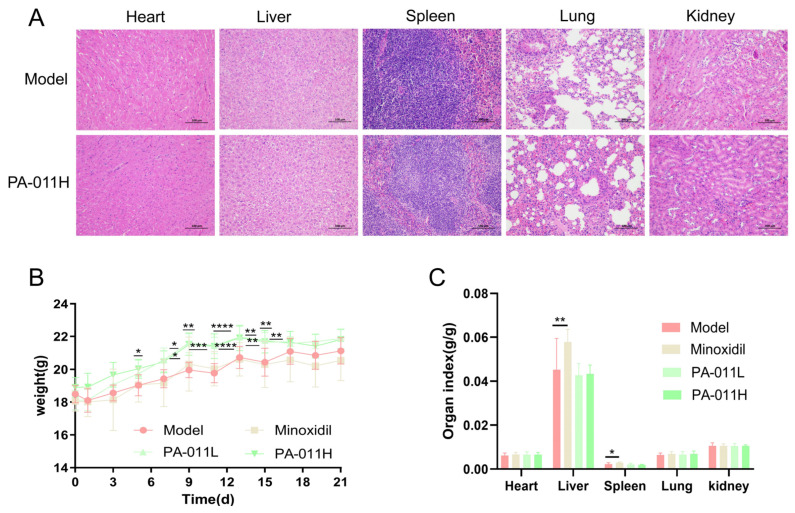
Safety evaluation of PA-011. (**A**) HE staining of pathological tissue sections of major organs (heart, liver, spleen, lung, kidney) (*n =* 4, Scale bars = 100 μm); (**B**) body weight changes during PA-011 intervention (*n* = 11); (**C**) organ indices of major organs (heart, liver, spleen, lung, kidney) (*n* = 11). ANOVA was performed. * *p* < 0.05, ** *p* < 0.01, *** *p* < 0.001, **** *p* < 0.0001 vs. model group.

**Figure 5 cimb-47-00619-f005:**
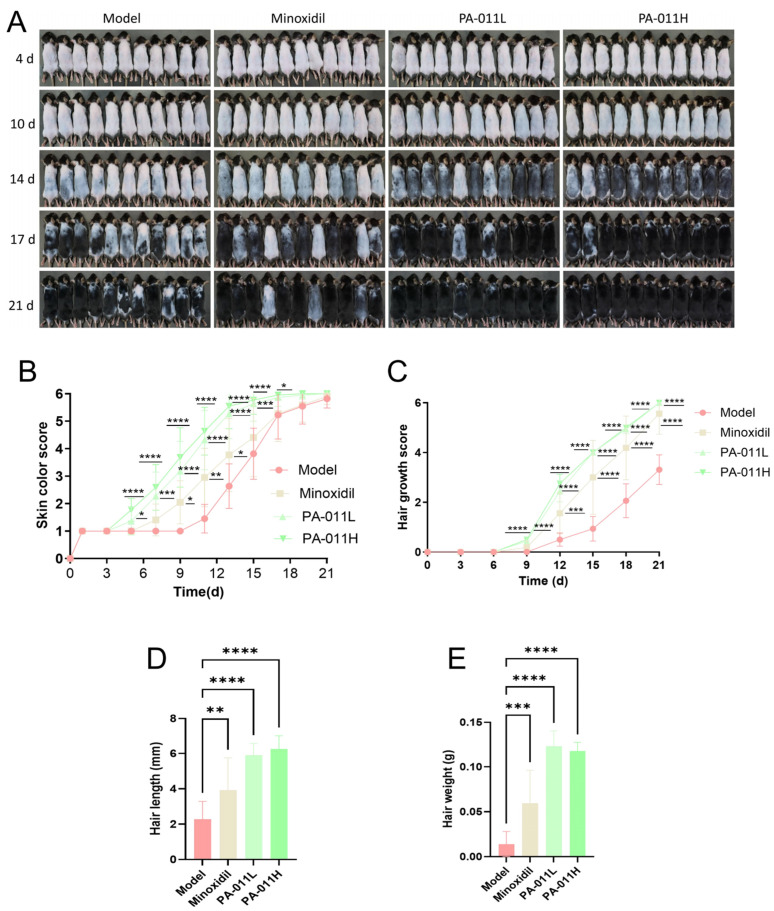
PA-011 promotes hair growth in mice. (**A**) Images of hair growth status at different time points (*n* = 11); (**B**) skin color scoring chart (*n* = 11); (**C**) hair growth scoring chart (*n* = 11); (**D**) hair length measurement chart (*n* = 11); (**E**) hair weight measurement chart (*n* = 8). ANOVA was performed. * *p* < 0.05, ** *p* < 0.01, *** *p* < 0.001, **** *p* < 0.0001 vs. model group.

**Figure 6 cimb-47-00619-f006:**
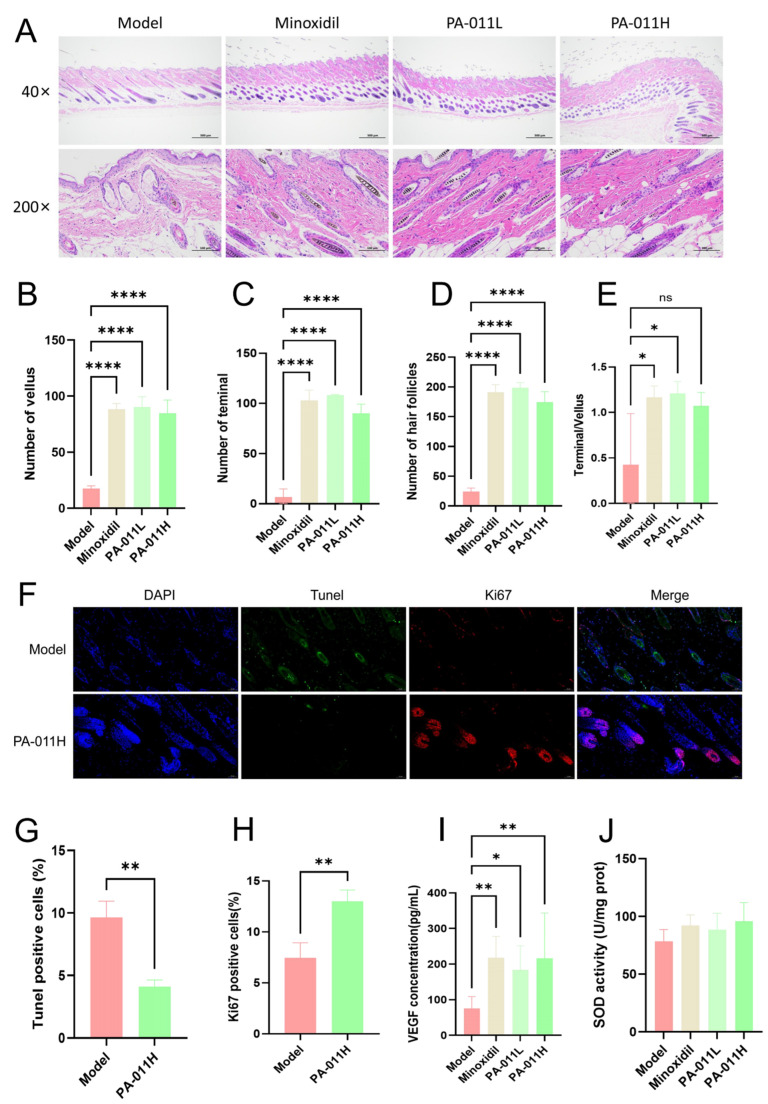
PA-011 alters hair growth status. (**A**) HE staining of skin pathological sections (*n* = 6, scale bars: 500 μm and 50 μm); (**B**) vellus follicle count; (**C**) terminal follicle count; (**D**) total HF count; (**E**) terminal-to-vellus follicle ratio; (**F**) Tunel and Ki67 staining of skin sections (*n* = 3, Scale bars: 50 μm); (**G**) proportion of Tunel-positive cells; (**H**) proportion of Ki67-positive cells; (**I**) VEGF content in skin tissue (*n* = 8); (**J**) SOD activity in skin tissue (*n* = 5). ANOVA was performed. * *p* < 0.05, ** *p* < 0.01, **** *p* < 0.0001 vs. model group.

**Figure 7 cimb-47-00619-f007:**
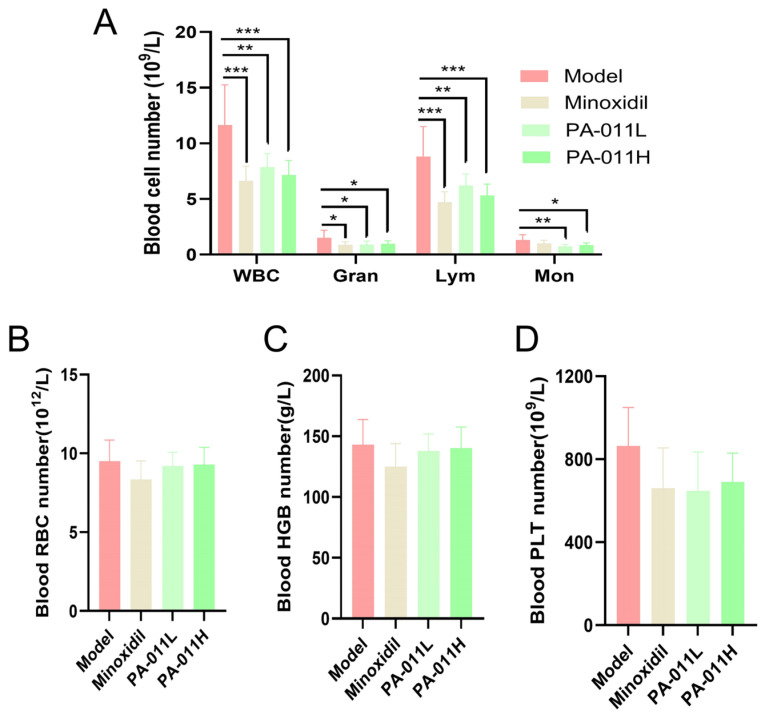
Effects of PA-011 on peripheral blood parameters in Ap mice (*n* = 8). (**A**) Counts of WBC, Gran, Mon, and Lym; (**B**) RBC count; (**C**) HGB concentration; (**D**) PLT count. ANOVA was performed. * *p* < 0.05, ** *p* < 0.01, *** *p* < 0.001 vs. model group.

**Figure 8 cimb-47-00619-f008:**
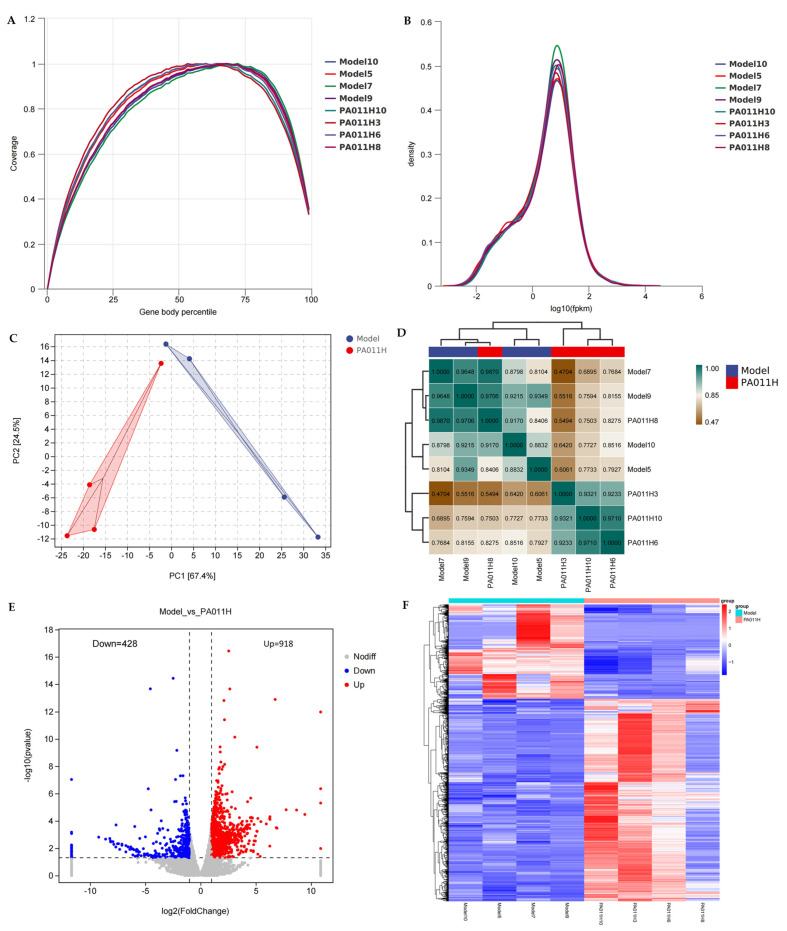
Differential transcriptomic analysis (*n* = 4). (**A**) Gene coverage uniformity; (**B**) FPKM density distribution; (**C**) PCA analysis; (**D**) correlation analysis; (**E**) volcano plot; (**F**) clustering analysis of differentially expressed genes; (**G**) genomic circle plot; (**H**) protein−protein interaction network analysis; (**I**) GO enrichment analysis; (**J**) KEGG enrichment analysis.

**Figure 9 cimb-47-00619-f009:**
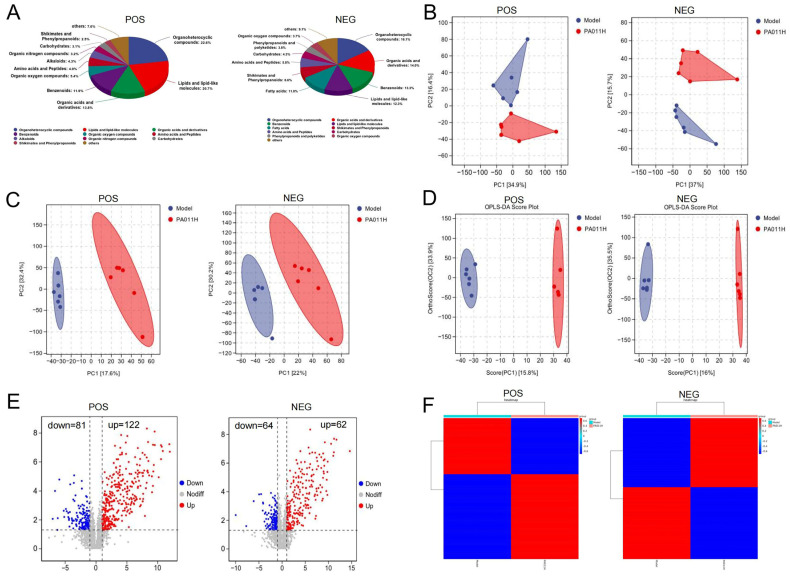
Non-targeted metabolomic differential analysis (*n* = 6). (**A**) Compositional analysis; (**B**) PCA analysis; (**C**) PLS−DA analysis; (**D**) OPLS−DA analysis; (**E**) volcano plot of differential metabolites; (**F**) clustering analysis of differential substances.

**Figure 10 cimb-47-00619-f010:**
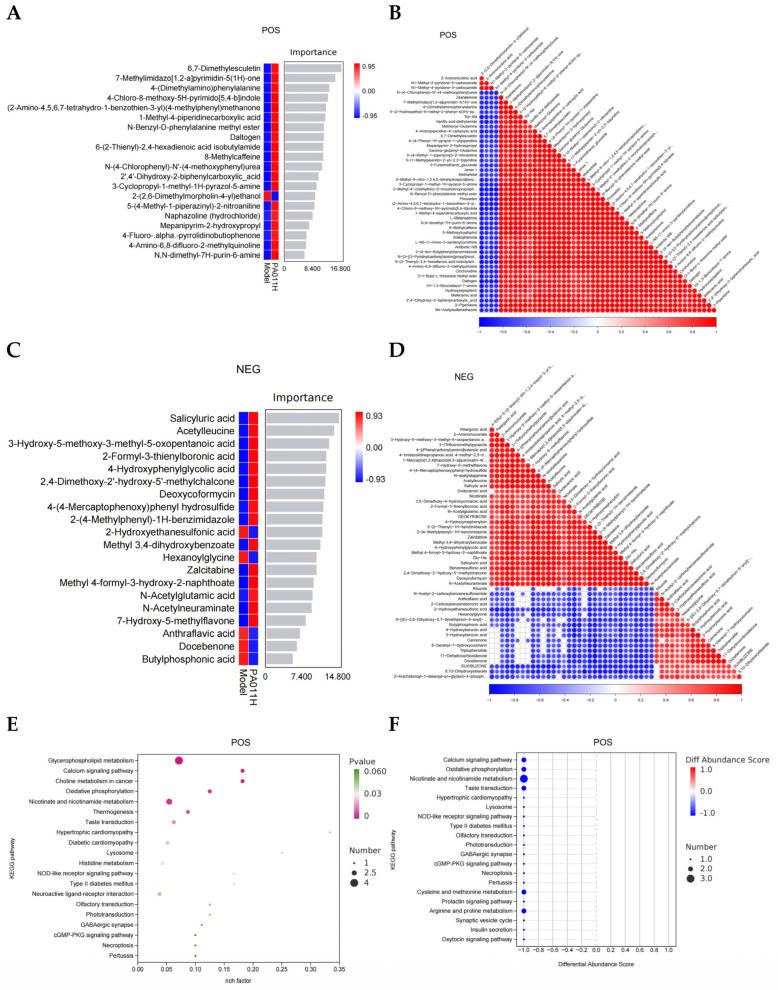
Functional analysis of key differential metabolites. (**A**) Scores of differential metabolites under positive ion mode; (**B**) correlation analysis of differential metabolites under positive ion mode; (**C**) scores of differential metabolites under negative ion mode; (**D**) correlation analysis of differential metabolites under negative ion mode; (**E**) KEGG enrichment analysis under positive ion mode; (**F**) abundance analysis of enriched pathways under positive ion mode; (**G**) KEGG enrichment analysis under negative ion mode; (**H**) abundance analysis of enriched pathways under negative ion mode.

**Figure 11 cimb-47-00619-f011:**
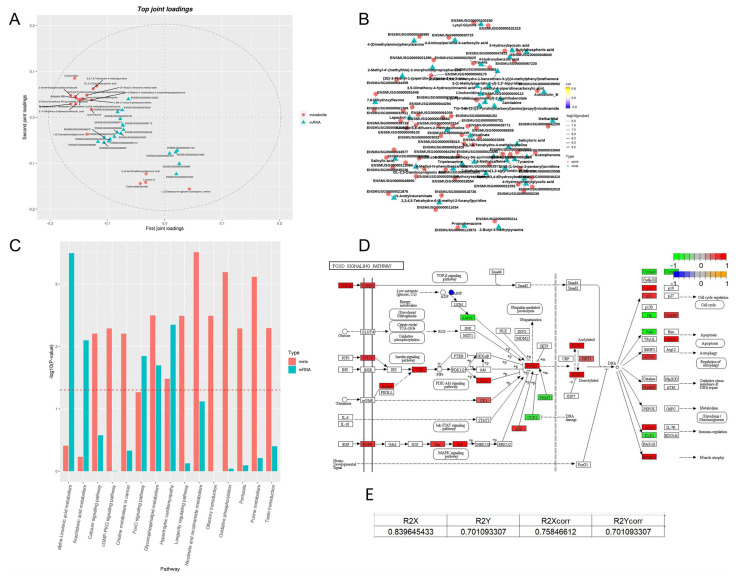
Integrated transcriptomics−metabolomics analysis. (**A**) Cross−omics correlation loading plot; (**B**) network diagram of top 100 correlation pairs with the smallest *p*-values; (**C**) bar chart of KEGG enrichment *p*-values; (**D**) enrichment of FoxO signaling pathway by differential metabolites and transcripts; (**E**) component proportions contribution assessment results.

**Figure 12 cimb-47-00619-f012:**
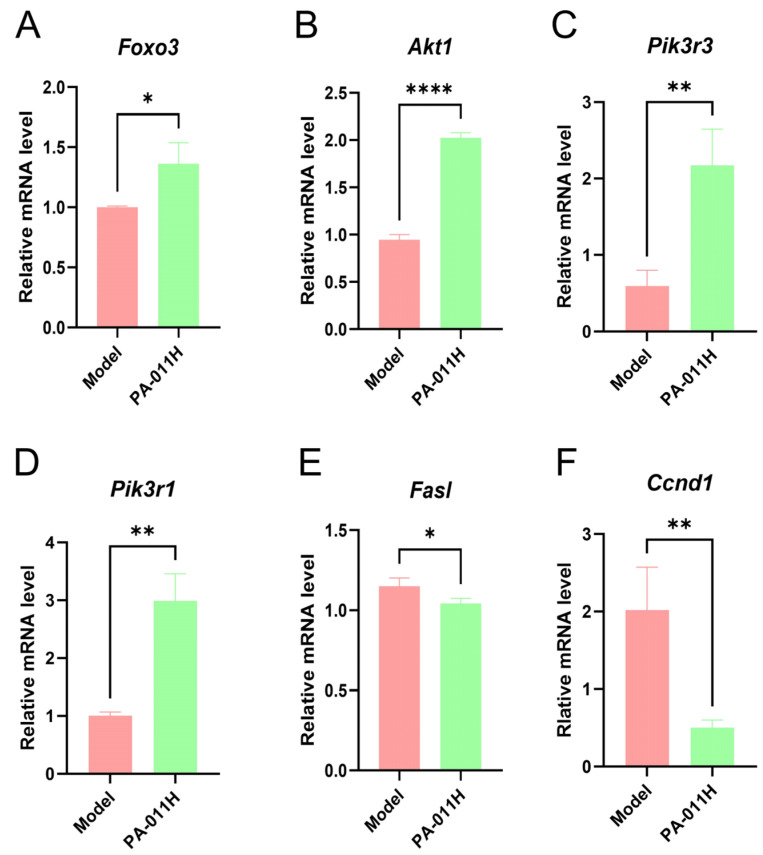
RT-qPCR analysis of skin tissues from Ap mice treated with PA-011 (*n* = 3). (**A**) *Foxo3* mRNA expression; (**B**) *Akt1* mRNA expression; (**C**) *Pik3r3* mRNA expression; (**D**) *Pik3r1* mRNA expression; (**E**) *Fasl* mRNA expression; (**F**) *Ccnd1* mRNA expression. Data were analyzed using independent samples *t*-tests. * *p* < 0.05, ** *p* < 0.01, **** *p* < 0.0001 vs. model group.

**Figure 13 cimb-47-00619-f013:**
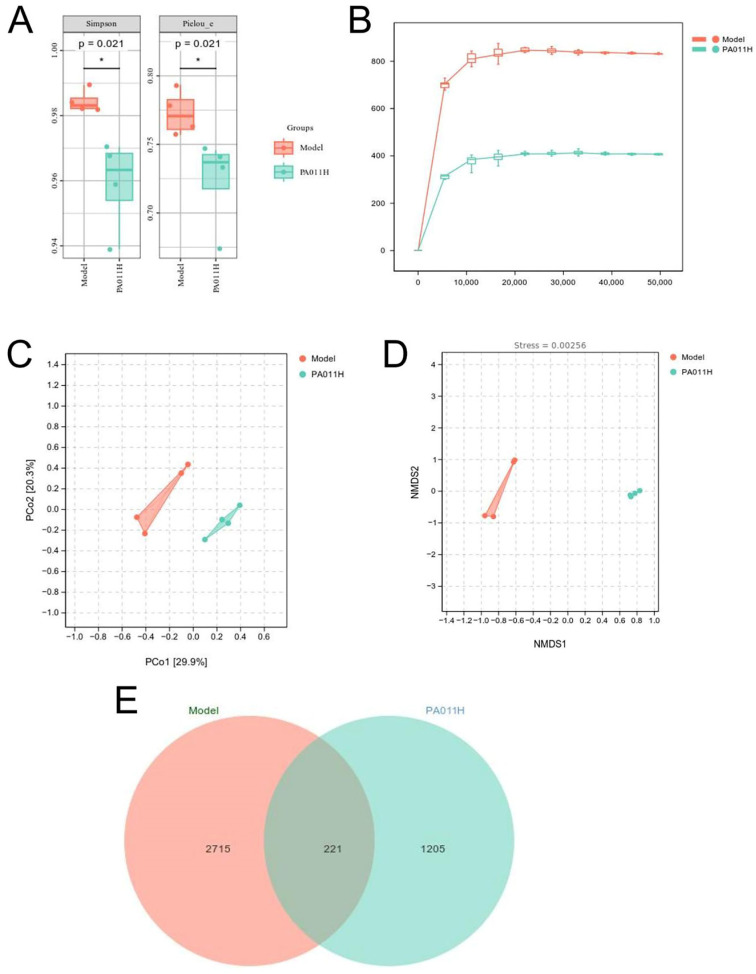
Skin microbiome diversity analysis (*n* = 4). (**A**) Alpha diversity indices; (**B**) rarefaction curves; (**C**) distance matrix and PCoA analysis; (**D**) NMDS analysis; (**E**) ASV Venn diagram. * *p* < 0.05.

**Figure 14 cimb-47-00619-f014:**
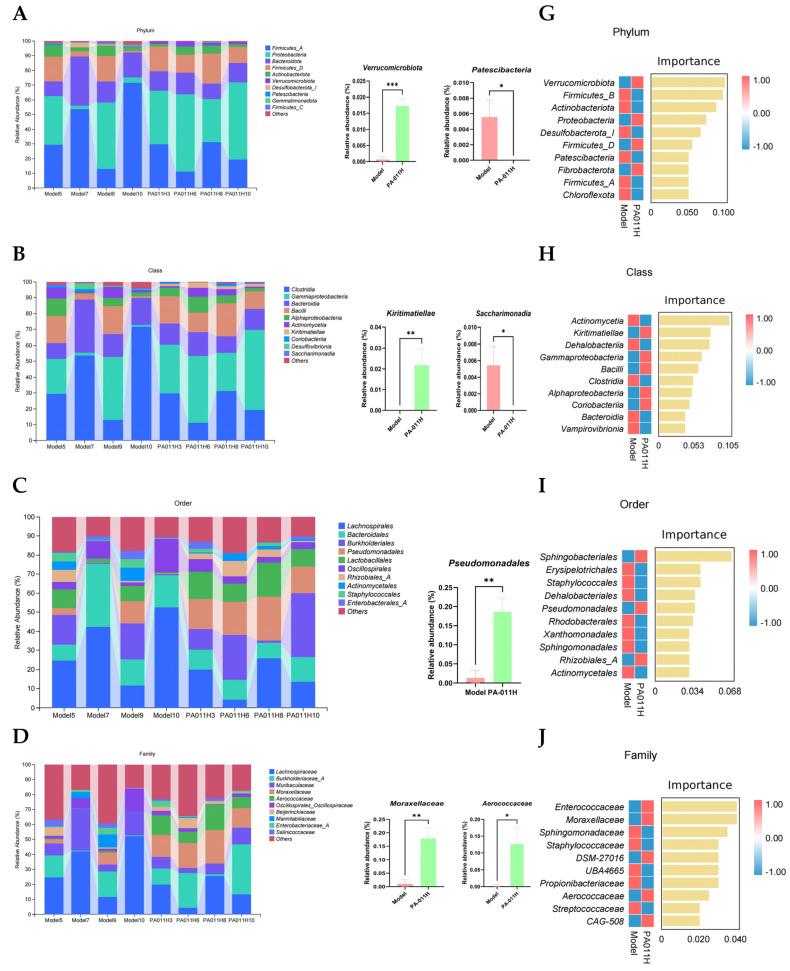
Effects of PA-011 on the microbiota of Ap mice. (**A**–**F**) Microbial composition of samples at six taxonomic levels (phylum, class, order, family, genus, species); (**G**–**L**) important microorganisms identified by machine learning analysis at six taxonomic levels (phylum, class, order, family, genus, species); (**M**) LDA bar chart; (**N**) taxonomic cladogram. * *p* < 0.05, ** *p* < 0.01, *** *p* < 0.001, **** *p* < 0.0001.

**Figure 15 cimb-47-00619-f015:**
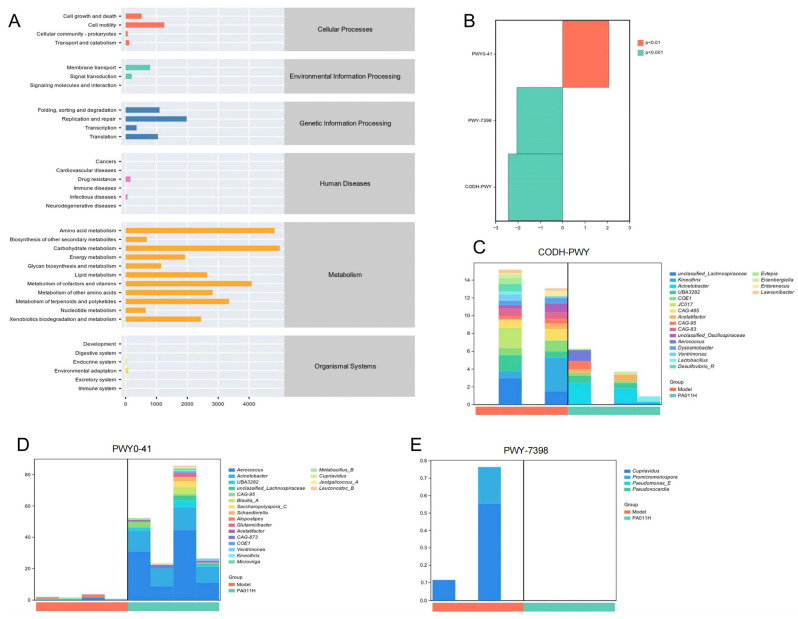
Predictive analysis of metabolic pathways in differential bacterial species. (**A**) Functional prediction of metabolic pathways; (**B**) significantly different pathways; (**C**) major microorganisms in the PWY0-41 pathway; (**D**) major microorganisms in the PWY-7398 pathway; (**E**) core species composing the CODH-PWY pathway.

**Table 1 cimb-47-00619-t001:** Primer sequence.

Name	Primer Direction	Primer Sequence (5′ to 3′)
*Foxo3*	Forward	GTGTTTGGACCTTCGTCTCTGA
	Reverse	GAGTGTCTGGTTGCCGTAGTGT
*Akt1*	Forward	GCTCTTCTTCCACCTGTCTCG
	Reverse	CGCAGAATGTCTTCATAGTGGC
*Pik3r3*	Forward	ATTGACTTTGAGGAAGGGAGGA
	Reverse	CTGTTGGAATCTGGATACTGGGT
*Pik3r1*	Forward	CACGGCGATTACACTCTTACACTA
	Reverse	CACTGGGTAGAGCAACTTCACATC
*Fasl*	Forward	CCTCTAAAGAAGAAGGACCACAACA
	Reverse	ACGGAGTTCTGCCAGTTCCTT
*Ccnd1*	Forward	AGGCGGATGAGAACAAGCAG
	Reverse	AAGAAAGTGCGTTGTGCGGTA
*Gapdh*	Forward	CCTCGTCCCGTAGACAAAATG
	Reverse	TGAGGTCAATGAAGGGGTCGT

**Table 2 cimb-47-00619-t002:** The top 16 most correlated small-molecule compounds.

Number	Name	ID	Molecular Formula	Structures
1	(R)-Methysticin	PA3	C_15_H_14_O_5_	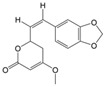
2	(S)-N-Methylcoclaurine	PA5	C_18_H_21_NO_3_	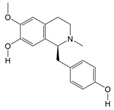
3	3-Hydroxyflavone	PA23	C_15_H_10_O_3_	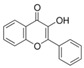
4	3-Methylindole	PA28	C_9_H_9_N	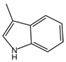
5	5,7-Dihydroxyflavone	PA39	C_15_H_10_O_4_	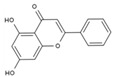
6	17a-Estradiol	PA49	C_18_H_24_O_2_	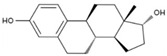
7	all-trans-Retinoic acid	PA51	C_20_H_28_O_2_	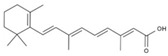
8	Dehydroepiandrosterone	PA66	C_19_H_28_O_2_	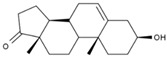
9	Dodecanedioic acid	PA72	C_12_H_22_O_4_	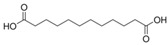
10	Epiandrosterone	PA74	C_19_H_30_O_2_	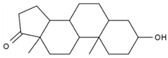
11	Exemestane	PA76	C_20_H_24_O_2_	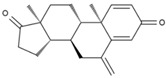
12	Hydroxyphenyllactic acid	PA85	C_9_H_10_O_4_	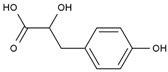
13	Methyl jasmonate	PA104	C_13_H_20_O_3_	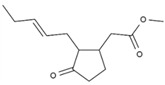
14	Oleic acid	PA123	C_18_H_34_O_2_	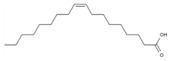
15	Palmitoleic acid	PA125	C_16_H_30_O_2_	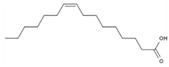
16	Sotalol	PA142	C_12_H_20_N_2_O_3_S	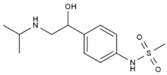

**Table 3 cimb-47-00619-t003:** The top 10 polypeptide sequences with degree value.

Number	Peptide	ID	Charge
1	FQQRPQPQPQPQPQ	peptide13	1
2	GGGAGGGAGGFGGGAGGGYR	peptide18	1
3	FYGVVRAP	peptide70	1
4	TPFYLR	Peptide76	1
5	FGGANR	peptide81	1
6	YAPR	Peptide86	1
7	SSFGPR	Peptide96	1
8	FGGGGAGGFGGGAGGR	Peptide97	1
9	AGGGFGGGSGGFGGRSP	peptide121	1
10	RYPYAPR	peptide129	2

## Data Availability

All the data are uploaded in the Mendeley Data database; the data set name is the same as the article title, DOI: 10.17632/x64tw97rc6.1.

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
