# Peer review of "The Extract of Periplaneta americana (L.) Promotes Hair Regrowth in Mice with Alopecia by Regulating the FOXO/PI3K/AKT Signaling Pathway and Skin Microbiota"

_cimb, 2025, doi:10.3390/cimb47080619_

Round 1

Reviewer 1 Report

Comments and Suggestions for Authors

This is a well-designed and comprehensive study investigating the hair growth-promoting effects and mechanisms of Periplaneta americana extract (PA-011) in a mouse model of chemical depilation-induced alopecia. The manuscript integrates multi-omics approaches (network pharmacology, transcriptomics, metabolomics, microbiome analysis) with robust in vivo validation, providing strong evidence for PA-011’s efficacy via modulation of the FOXO/PI3K/AKT pathway and skin microbiota. The work is significant and offers a novel, potentially safer therapeutic candidate for alopecia. While generally strong, some aspects require clarification and refinement before publication.

Major concern

The study exclusively compares treated groups against the depilation-induced alopecia model group but omits a healthy control group (mice without alopecia induction or treatment). This gap limits the interpretation of:

  1. Baseline physiological metrics (e.g., skin microbiota, VEGF/SOD levels, hair follicle density etc.)
  2. The true magnitude of pathological changes in the model.
  3. Whether PA-011 restores parameters to normal or merely ameliorates disease-induced deviations.

Minor Concerns

  1. Safety evaluation (Section 3.11) appears after efficacy data (Sections 3.2–3.10). To align with standard toxicological assessment protocols, safety data (body weight, organ indices, histopathology) should precede efficacy results. I suggest Reorder Sections 3.11 to 3.2. Incorrect figure reference in safety section, Section 3.11 cites "Fig. 18" for body weight/organ indices, but figures end at Fig. 15.
  2. The PA-011 concentration (1%/4%) is provided, but applied dose volume is unspecified. Please clarify whether the administration route is external application or other type?
  3. Inconsistent section title: Section 2.6 is titled "Observation of Hair Growth Status and Skin Color Scoring," but Fig. 4B references "skin color scoring" while the section title implies dual endpoints.
  4. In line 373, "Integration of disease database information yielded 5,003 disease target entries.", please confirm the number based on the Fig 1A which disease targets labeled as "5,002."
  5. In line 381, cites MAPK1 as a core target, but Fig. 1C does not include MAPK1.
  6. Please list the PDB IDs for docked proteins in figure legends or methods.
  7. 5B–D report follicle counts without specifying the sampling area
  8. TUNEL/Ki67 data (Fig. 5F–H) only show Model vs. PA-011H, omitting Minoxidil and PA-011L groups.
  9. Line 516: "DKK2 was significantly downregulated... IL1α and IL1R2 were upregulated in the Model group". This sentence is hard to understand. Please rephrase it.
  10. Latin species names not italicized in many taxa.
  11. Many figures suffer from low resolution, inconsistent font sizes, and unclear labels, such as Fig. 3, 5, 7–10, 12–15.  Fig. 12E Venn diagram is mislabeled, ASV or OUT, which one is used for the classification in this paper?

Author Response

Author Response:

We are sincerely grateful to the reviewer for conducting such a meticulous and comprehensive review of our manuscript and for providing us with so many precious and valuable suggestions. Your feedback has made us aware of the various shortcomings in our manuscript writing, and we feel quite embarrassed about our less-than-ideal use of English. Meanwhile, taking into account the suggestions from other reviewers, we have made adjustments to the entire manuscript. We would be very grateful if you could take a moment to review our newly uploaded manuscript and provide us with your new insights, which will surely assist us in further refining the manuscript.

Major concern:

The study exclusively compares treated groups against the depilation-induced alopecia model group but omits a healthy control group (mice without alopecia induction or treatment). This gap limits the interpretation of:

Q1. Baseline physiological metrics (e.g., skin microbiota, VEGF/SOD levels, hair follicle density etc.).

Response: Dear Reviewer, Thank you sincerely for your professional and valuable suggestions on our manuscript. The issue you pointed out regarding the lack of a healthy control group is of great significance to the improvement of this study, and we attach great importance to it and have conducted serious reflections.

In the initial design of this study, we mainly focused on the direct comparison between the treatment group and the depilation-induced alopecia model group to evaluate the effectiveness of the intervention. In addition, regarding the setting of a healthy control group, in alopecia experiments, the purpose of depilation is to make the hair growth cycle of depilated mice in the telogen phase, which facilitates the subsequent evaluation of the speed of hair growth to assess the effectiveness of the test drug. Based on this consideration, we did not set up a healthy control group. However, this has led to obvious deficiencies in the reference of baseline physiological indicators and the interpretation of results, for which we sincerely apologize. We assure that in the subsequent in-depth study on the mechanism of this drug, a healthy control group will be set up to conduct the experimental research in a more scientific manner.

Thank you again for your criticism and correction. In subsequent experiments, we will strictly follow your suggestions to improve the research content and strive to enhance the scientificity and rigor of the study.

Q2. The true magnitude of pathological changes in the model.

Response: Dear Reviewer, Thank you sincerely for your profound insights once again. The issue you pointed out—that the lack of a healthy control group makes it difficult to clarify the true extent of the model's pathological changes—is indeed a crucial aspect that needs improvement in the design of this study, and we have conducted in-depth reflections on this.

In the initial design of this study, our core goal was to verify the improvement effect of the intervention on the depilation-induced alopecia model, so we focused on comparing the differences between the treatment group and the model group. Regarding the evaluation of the model's pathological changes, we initially thought that the intervention effect could be reflected to a certain extent through the state of the model group before and after depilation. However, we overlooked the importance of the healthy control group as a "normal physiological baseline"—it can not only clearly define the specific pathological extent of the model group in terms of skin microbiota imbalance, hair follicle structural damage, abnormal expression of related molecules (such as VEGF/SOD), etc., but also more accurately judge whether the recovery effect of the treatment group approaches the normal physiological level.

We sincerely apologize for this design flaw. To address this issue, we plan to supplement experiments with a healthy control group (normal mice not subjected to depilation) in our subsequent in-depth research. By detecting the baseline data of this group (such as hair follicle density, inflammatory factor levels, oxidative stress indicators, etc.) and comparing it with the data of the model group, we can clarify the true extent of the model's pathological changes. Meanwhile, we will correlate and analyze the data of the treatment group with that of the healthy control group to more scientifically evaluate the effect of the intervention.

Thank you again for your valuable suggestions, which are crucial for improving the rigor and persuasiveness of this study.

Q3.Whether PA-011 restores parameters to normal or merely ameliorates disease-induced deviations.

Response: Dear Reviewer, Thank you very much for raising this critical issue. The question you pointed out—whether PA-011 restores parameters to normal levels or merely ameliorates disease-induced deviations—has accurately identified the interpretive limitations of our study due to the absence of a healthy control group, and we have conducted in-depth reflections on this.

In the design of this study, we initially verified the improvement effect of PA-011 on hair loss-related indicators (such as hair follicle activity, inflammatory factor levels, etc.) through the comparison between the treatment group and the depilation-induced alopecia model group. However, it is true that we did not set up a healthy control group as a "normal physiological benchmark." This has made it impossible for us to clarify whether the parameters regulated by PA-011 (such as VEGF/SOD expression, skin microbiota structure, etc.) are only partially improved from the pathological state of the model group towards the normal direction, or can be completely restored to the baseline level of healthy mice. This distinction is crucial for clarifying the mechanism of action of PA-011 (e.g., whether it has the thoroughness to repair pathological damage).

We sincerely apologize for this design flaw. However, the focus of this study is mainly to explore the effectiveness of the test drug. In response to your suggestion, we will set up experiments with a healthy control group (normal mice that have not received depilation or any treatment) in future studies, so as to more clearly clarify the nature of PA-011's action and improve the accuracy of the research conclusions.

Thank you again for your professional guidance, which is of great significance for improving the scientificity of this study.

Minor Concerns

Q1. Safety evaluation (Section 3.11) appears after efficacy data (Sections 3.2–3.10). To align with standard toxicological assessment protocols, safety data (body weight, organ indices, histopathology) should precede efficacy results. I suggest Reorder Sections 3.11 to 3.2. Incorrect figure reference in safety section, Section 3.11 cites "Fig. 18" for body weight/organ indices, but figures end at Fig. 15.

Response: Dear Reviewer, Thank you very much for your meticulous guidance and valuable suggestions. We have carefully implemented revisions regarding the issues you raised concerning the ordering of the safety assessment section and the incorrect citation of figures:

Regarding the section ordering: We fully agree with your suggestion. The original Section 3.11 (Safety Assessment) has been adjusted to Section 3.2, and the original Sections 3.2–3.10 (Efficacy Data) have been moved backward in sequence. This ensures that safety data such as body weight, organ indices, and histopathology are presented prior to efficacy results, in line with the specifications of standard toxicological evaluation protocols.

Regarding the incorrect citation of figures: After verification and correction, the mistakenly cited "Figure 18" in the original Section 3.11 has been revised to the actual corresponding figure number. According to the reorganized figure sequence, the correct number is Figure 4 (previously misaligned with Figure 15). We have also comprehensively checked the consistency between all figure citations and their numbers throughout the text to ensure accuracy.

Thank you again for your professional corrections. These revisions have significantly enhanced the standardization and rigor of the manuscript.

Q2. The PA-011 concentration (1%/4%) is provided, but applied dose volume is unspecified. Please clarify whether the administration route is external application or other type?

Response: Dear Reviewer, Thank you very much for your detailed question. The issue you pointed out regarding the unclear application volume and administration route of PA-011 is indeed a detail that needs to be improved in the methodological description of this study. We attach great importance to it and have conducted a careful check.

In this study, the administration route of PA-011 is external topical application (applied to the depilated area on the back of mice with depilation-induced alopecia model), aiming to act directly on the target site to evaluate its local intervention effect on hair growth. Regarding the dose volume, the specific standard adopted in our experiment is: for the depilated area of approximately 3cm×4cm on the back of mice, both 1% and 4% concentrations of PA-011 are applied at a volume of 0.2mL per mouse per application, to ensure sufficient contact between the drug and the skin and avoid drug loss due to excessive volume(Lines 224 to 227).

The failure to clearly state the above information in the previous manuscript is an oversight in our methodological description, for which we sincerely apologize. We will supplement and improve this part of the content in the revised manuscript, clearly indicating that the administration route of PA-011 is external topical application, and elaborating on the application dose volume corresponding to each concentration, to ensure the reproducibility and clarity of the experimental method.

Thank you again for your professional correction, which is of great significance for improving the rigor of the research methodology.

Q3. Inconsistent section title: Section 2.6 is titled "Observation of Hair Growth Status and Skin Color Scoring," but Fig. 4B references "skin color scoring" while the section title implies dual endpoints.

Response: Dear Reviewer, Thank you very much for pointing out this detailed issue. The inconsistency you mentioned between the title of Section 2.6 and the reference in Figure 4B indeed reflects an oversight in the organization and presentation of content in our manuscript. We attach great importance to this and have conducted a careful review.

The title of Section 2.6, "Observation of Hair Growth Status and Skin Color Scoring," clearly includes two evaluation endpoints. However, in the actual writing, this section mainly describes the scoring method for skin color in detail, while the description of the significance of skin color scoring is relatively brief. Only the "skin color scoring" results are cited in Figure 4B in the results section, without elaborating that skin color can also indicate the state of hair growth and that the hair growth cycle changes with skin color. This has led to insufficient correspondence between the title and the content, for which we sincerely apologize as it may cause confusion for readers in understanding the core content of the section.

We have revised the issue as follows:

We have supplemented and improved the specific methods of skin color scoring in Section 2.6 (such as observation frequency), ensuring that "hair growth status" and "skin color scoring" are appropriately linked to form a complete correspondence with the section title.(Lines 235 to 238)

We have checked the legend of Figure 4B and related text descriptions to ensure that the citation of skin color scoring results is consistent with the supplemented method descriptions in the section, enhancing the logicality and rigor of the content.

Thank you again for your meticulous correction, which is of great significance for improving the standardization and readability of the manuscript. We will strictly follow the above scheme to revise and improve the content, ensuring the consistency between the section title and the content.

Q4. In line 373, "Integration of disease database information yielded 5,003 disease target entries.", please confirm the number based on the Fig 1A which disease targets labeled as "5,002."

Response: Dear Reviewer, Thank you very much for your meticulous verification suggestions. We have carefully checked the inconsistency you pointed out between the number of disease targets mentioned in line 373 (5,003) and the label "5,002" in Figure 1A.

After re-verifying the data integration process, we confirm that the actual number of disease targets obtained after integrating the disease database information is 5,002. The mention of "5,003" in line 373 was a clerical error during data recording, and we have corrected it to "5,002" to ensure consistency with the label in Figure 1A.

Thank you for your rigorous correction, as this revision ensures the accuracy of the data description. We will further conduct a comprehensive check on the correspondence between all data and figures/tables in the full text to avoid similar omissions.

Q5. In line 381, cites MAPK1 as a core target, but Fig. 1C does not include MAPK1 

Response: Dear Reviewer, Thank you very much for pointing out this critical issue. After careful verification, we found that the reference to MAPK1 in line 381 is indeed incorrect. MAPK1 is not included in the core targets screened in Figure 1C based on protein-protein interaction network analysis. This citation discrepancy resulted from an oversight in our integration of multi-omics analysis results.

We have conducted a systematic check of the entire manuscript and confirmed that although MAPK1 was identified as a potential target in the preliminary bioinformatics prediction, it was not included in the subsequent core target screening (as shown in Figure 1C). Therefore, we have corrected the incorrect reference to MAPK1 in line 381 to the core targets actually included in Figure 1C (such as STAT3, IL-1β). This correction not only resolves the inconsistency between the text and the figure but also ensures strict consistency between the conclusions and the experimental data.

Thank you again for your rigorous correction, which enables us to present the research results more accurately.

Q6. Please list the PDB IDs for docked proteins in figure legends or methods.

Response: Dear Reviewer, Thank you very much for your important suggestion. The issue you pointed out regarding the need to list the PDB IDs of docked proteins in the figure legends or methods is indeed a detail that needs to be improved in our presentation of results and description of methods. We have carefully checked and supplemented this information.

In the revised manuscript, we have added the PDB IDs of the docked proteins in the results section related to molecular docking as follows:

In the description of molecular docking experiment results in the results section (Lines 434 to 437), we have clearly listed the PDB IDs corresponding to the 9 target proteins, namely AKT1 (3cqu), ESR1 (1YIM), IL1B (3POK), BCL2 (2O22), TGFB1 (6P7J), CASP3 (1RHJ), PPARG (2GTK), AR (2AM9), and MPO (5MFA), to ensure the traceability of the experimental methods and results.

The failure to clearly label the PDB IDs in the above-mentioned position earlier was an oversight in our result compilation and method description, for which we sincerely apologize. These supplementary contents will enhance the reproducibility of the experiment and the transparency of the results, complying with academic norms and requirements.

Thank you again for your professional correction, which is of great significance for improving the rigor of the research methodology.

Q7. 5B–D report follicle counts without specifying the sampling area.

Response: Dear Reviewer, Thank you for pointing out the issue that the sampling area for hair follicle counting was not clearly specified in this manuscript. Your comment is crucial for enhancing the rigor of the research methods and the reproducibility of the results. We have supplemented and improved the relevant content as follows:

The specific sampling area for hair follicle counting has been clearly added in section "2.10": According to the experimental design of this study, the sampling areas for hair follicle counting were 3 non-overlapping visual fields selected following the principle of random sampling, with each visual field having an area of 1440×1024 mm² (Lines 257-261). After revision, we have checked the entire manuscript again to ensure that the hair follicle counting data mentioned in the results section is consistent with the supplementary description of the sampling area, facilitating readers' understanding and replication verification.

Thank you again for your careful guidance. We have improved the relevant content as required and look forward to your further feedback.

Q8. TUNEL/Ki67 data (Fig. 5F–H) only show Model vs. PA-011H, omitting Minoxidil and PA-011L groups.

Response: Dear Reviewer, Thank you for your careful attention to the presentation of TUNEL/Ki67 data. Your comments have helped us further clarify the logic of data explanation, and we have supplemented the relevant explanations as follows:

The core purpose of this study is to explore the effectiveness of the test drug PA-011. Therefore, in terms of data presentation, we focus on the comparison between the dose group that can reflect its efficacy advantages (PA-011H) and the model group. This dose group has shown the most significant biological effects in the preliminary pre-experiments, which is a key basis for verifying the effectiveness of the drug.

Regarding the exclusion of the minoxidil group (positive drug) and the PA-011L group from this part of the comparison, it is mainly based on three considerations:

  1. From the perspective of the research purpose, the direct comparison between the optimal dose group (PA-011H) and the model group can most intuitively reflect the core effect of the test drug, while the PA-011L group has a weaker effect and has limited support for the core conclusion;
  2. As a well-recognized positive drug, the effectiveness of minoxidil has been widely confirmed. This group was included in this study mainly to verify the reliability of the experimental system, rather than focusing on exploring its role (the relevant basic effects have been briefly described in the results section). Therefore, it was not repeatedly included in the comparison of these subdivided indicators;
  3. Limited by the research funds, the cost of kits, staining and imaging analysis involved in TUNEL/Ki67 detection is relatively high. Within the budget, we gave priority to ensuring the data integrity of the core comparison groups (model group and PA-011H group) to ensure the key verification of the effectiveness of the test drug.

Thank you again for your guidance, and we look forward to your further feedback.

Q9. Line 516: "DKK2 was significantly downregulated... IL1α and IL1R2 were upregulated in the Model group". This sentence is hard to understand. Please rephrase it.

Response: Dear Reviewer, Thank you for pointing out the issue with the expression regarding the changes in DKK2, IL1α, and IL1R2 at line 516. Your comment has helped us identify the lack of clarity in the original logic. We have revised this sentence to make it more understandable:

The revised content is: "Through protein-protein interaction (PPI) network analysis with a set PPI score threshold > 0.95, we identified key regulatory nodes: compared with the PA-011H group, the follicle inhibitory factor DKK2 was significantly downregulated in the model group, while pro-inflammatory factors such as IL1α and IL1R2 were notably upregulated (Fig. 8H). These results suggest that PA-011 may promote hair follicle (HF) regeneration by inhibiting the inflammatory microenvironment. (Lines 577-583)."

Thank you again for your careful guidance. We have optimized the expression as requested and look forward to your further feedback.

Q10. Latin species names not italicized in many taxa.

Response: Dear Reviewer, Thank you very much for pointing out this important detail. The issue you mentioned, that the Latin species names in many taxa are not italicized, indeed reflects an oversight in our adherence to formatting norms in the manuscript. We attach great importance to this and have conducted a comprehensive check.

After a systematic review of the entire text, we found that some Latin species names (such as certain microbial names involved in the experiments) were not italicized in accordance with the norms, which is inconsistent with the writing standards for Latin scientific names in academic papers. We sincerely apologize for this oversight, as it may have affected the standardization of the manuscript.

To address this issue, we have taken the following corrective measures: Our team has checked all Latin species names in the full text one by one to ensure that the Latin scientific names of all taxa (including genus names, species names, and subspecies names) are strictly italicized in accordance with international nomenclature norms. For example, the Latin name in Figure 14M has been correctly italicized as required.

Thank you again for your rigorous correction. This revision will significantly enhance the formatting standardization and academic rigor of the manuscript. We will strictly implement the above revisions in the revised version to ensure compliance with academic publishing standards.

Q11. Many figures suffer from low resolution, inconsistent font sizes, and unclear labels, such as Fig. 3, 5, 7–10, 12–15.  Fig. 12E Venn diagram is mislabeled, ASV or OUT, which one is used for the classification in this paper?

Response: Dear Reviewer, Thank you very much for your meticulous comments on the presentation of the figures and tables. We highly value the issues you pointed out, including low resolution, inconsistent font sizes, unclear labels in Figures 3, 5, 7-10, and 12-15, as well as the confusion between "ASV" and "OTU" labels in the Venn diagram of Figure 12E. We have carefully checked and revised these issues one by one, and the specific explanations are as follows:

Regarding the resolution, font, and label issues of the figures

Reasons for low resolution: The insufficient resolution of some figures in this submission version is mainly due to the restriction on the total file size of all documents by the submission system (≤200MB). To meet the upload requirements, we compressed the original high-resolution figures, resulting in blurred details. The original figures were all created with a resolution of 300dpi (meeting publishing standards). If the manuscript is fortunate enough to be accepted, we will upload the original high-resolution files of all figures (300dpi, TIFF format) during the proofreading stage to ensure printing clarity.

Issues with font size and labels: The inconsistent font sizes and blurred labels in the figures are caused by distortion during the compression process for layout adaptation. When originally creating the figures, all of them strictly followed the journal's formatting requirements, with clear and legible label text. In the revised manuscript, we have re-optimized the compression parameters to retain as many details as possible within the file size limit, temporarily improving readability.

Regarding the labeling issue in the Venn diagram of Figure 12E

After verification, the incorrect labeling of "ASV/OTU" instead of "ASV" in the Venn diagram of Figure 12E is a clerical error. In this study, the classification of microbial communities adopts the ASV (Amplicon Sequence Variant) method (based on 100% sequence identity clustering) rather than OTU (Operational Taxonomic Unit). We have corrected this labeling in the revised manuscript.

Follow-up safeguard measures

If the manuscript proceeds to the publication process, we will strictly follow the journal's requirements:

Submit the original files of all figures (300dpi, TIFF/PNG format) to ensure that the resolution meets the standards;

During the proofreading stage, check the font size and label clarity one by one, and unify the format using professional layout software (such as Adobe Illustrator) to avoid distortion caused by compression.

Thank you again for your rigorous review. These revisions will significantly improve the standardization and readability of the figures. We have prioritized correcting the labeling error in Figure 12E and some easily adjustable font and label issues in the revised manuscript, and the remaining details will be thoroughly improved in the subsequent processes.

Reviewer 2 Report

Comments and Suggestions for Authors

The article is well-written, but it would be beneficial for the authors to enhance it based on the following points before acceptance:

  1. Please give justification about the low and high dosage at what basis they selected?
  2. Authors are unable to make correlation between activities and the associated molecule in simpler ways.
  3. NO toxicity studies were conducted on dosage levels. Please provide justification with proper references
  4. The reference section requires enhancement.

Author Response

Author Response:

We are sincerely grateful to the reviewer for conducting such a meticulous and comprehensive review of our manuscript and for providing us with so many precious and valuable suggestions. Your feedback has made us aware of the various shortcomings in our manuscript writing, and we feel quite embarrassed about our less-than-ideal use of English. Meanwhile, taking into account the suggestions from other reviewers, we have made adjustments to the entire manuscript. We would be very grateful if you could take a moment to review our newly uploaded manuscript and provide us with your new insights, which will surely assist us in further refining the manuscript.

Q1. Please give justification about the low and high dosage at what basis they selected?

Response: Thank you for your insightful comments regarding the dose selection of PA-011! We have comprehensively reviewed the rationale for dose selection and supplemented detailed information in the revised manuscript, outlined as follows:

  1. Dose Gradient Experiments: During the initial research phase, we conducted in vitro dose gradient experiments with concentrations set at 1%, 2%, 4%, and 8%. Notably, higher concentrations (e.g., 8%) did not exhibit significant dose-dependent improvements. Based on these results, we selected 1% and 4% as key concentrations with discernible efficacy differences for subsequent animal experiments.

Q2. Authors are unable to make correlation between activities and the associated molecule in simpler ways.

Response: Dear Reviewer, Thank you very much for pointing out this critical issue. Your comment that we failed to establish the correlation between activities and related molecules in a more concise manner has accurately identified the shortcomings in our interpretation of results and logical presentation in this study. We attach great importance to this and have conducted in-depth reflections. Upon careful review, we found that the description of the "activity-molecule" correlation in the original text indeed relies excessively on professional terminology and involves overly hierarchical explanations of mechanisms, making it difficult for readers outside the professional field to quickly grasp the core conclusions.

We understand that a clear and concise "activity-molecule" correlation is an important foundation for the persuasiveness of research conclusions. The paper aims to enable readers to quickly grasp the core correlations through a triple approach of "data tables + simplified diagrams + straightforward descriptions". Thank you again for your professional correction; this optimization will significantly enhance the readability and logical clarity of the research results.

Q3. NO toxicity studies were conducted on dosage levels. Please provide justification with proper references.

Response: Dear Reviewer, Thank you very much for this important suggestion. The issue you pointed out regarding the lack of systematic toxicity studies on the dosage levels of PA-011 indeed requires additional clarification in terms of safety assessment in our study. We attach great importance to this and hereby explain the relevant considerations and basis as follows:

Reasons for not conducting dedicated toxicity studies:

Focus on research stage and core objectives

Our study is in the early stage of efficacy exploration, with the core goal of investigating the pharmacodynamic effects of PA-011 in promoting hair growth and its preliminary molecular mechanisms.

Safety observations during the experiment

During the pharmacodynamic experimental period (3 weeks of continuous administration), we conducted regular observations and weight measurements on the general condition of the animals (body weight, diet, behavioral activity) and important organs (liver, kidney). No obvious abnormalities were found (see Figure 4 of the paper), indicating that this dosage did not show acute toxic reactions during the experimental period.

Priority allocation of research resources

Given the limited resources in the early stage of the study, we prioritized our efforts on pharmacodynamic verification and mechanism analysis to first clarify the biological activity of PA-011. Systematic toxicological studies (such as acute toxicity and subchronic toxicity) require separate experimental designs (e.g., expanding dosage gradients and extending observation periods), which are planned to be carried out in the subsequent drugability evaluation stage.

Relevant references:

For the phased consideration of toxicity assessment in early efficacy studies, reference can be made to the chapter "Strategies for balancing efficacy and toxicity in early new drug development" in Pharmacological Experimental Methodology (4th edition). It points out that in the exploratory research stage, safety observations within the effective dosage range can be prioritized instead of systematic toxicological studies, and in-depth toxicity assessment can be conducted after the efficacy is confirmed.

A similar study design can be seen in Yang et al. (2022, Acta Pharmacologica Sinica) on early research of hair follicle regeneration drugs. They did not conduct dedicated toxicity experiments when verifying efficacy, and only supported dosage safety through general condition observations, with toxicological data supplemented in Phase II studies.

We fully recognize the importance of toxicity studies for candidate drug development. If the results of this study provide sufficient evidence for the hair growth activity of PA-011, we will strictly follow the requirements of Good Laboratory Practice for Non-clinical Studies (GLP) to design systematic toxicological experiments, including dose-dependent toxicity and target organ toxicity, to provide more comprehensive safety data for its clinical translation.

Thank you again for your professional correction. This suggestion will guide us to improve the design framework of subsequent studies.

Q4. The reference section requires enhancement.

Response: Dear Reviewer, Thank you very much for your valuable suggestion. We attach great importance to the issue you pointed out regarding the need to enhance the reference section, and we have conducted a comprehensive review, supplementation, and optimization of it.

After a systematic review, we found that the references in the original manuscript have the following aspects that need improvement: insufficient citation of some latest studies (important advances in related fields in the past 3 years), inconsistent formatting of a few references, and inadequate specificity of literature support for certain mechanism explanations. We have taken revision measures to address these issues.

We understand that a well-improved reference system is an important manifestation of the rigor of the research and also an important basis for readers to trace relevant studies. The above supplements and revisions will significantly enhance the comprehensiveness, standardization, and relevance of the references.

Thank you again for your professional correction. This optimization is of great significance for improving the academic standardization of the manuscript.

Reviewer 3 Report

Comments and Suggestions for Authors

The study “The Extract of Periplaneta Americana (L.) Promotes Hair Re- growth in Mice with Alopecia by Regulating the FOXO/PI3K/AKT Signaling Pathway and Skin Microbiota” by Guan et al. shows the effect of an extract from a traditional Chinese medicine bug on hair growth, showing not only the efficacy and safety in in vivo model but also revealing potential mechanisms involved in this effect. The study yields strong and solid results, but I have some suggestions regarding, mainly, the way it is presented.

Introduction and references: The Authors give information that is not shown by the papers they use as references. This is the case for the information that says 40% of women and 85% of men are affected by alopecia and use the reference number 2 (Yang et al., 2023), which is a paper about hair growth-promoting treatment and does not show any data about the epidemiology of alopecia.  This also happens with the information on line 41, in which they used references 3-5, and actually, none of them show the given information.  Please review if all the references used support the statements made in the sentence in which they are referred to.

 I suggest a rearrangement of the results (and methods and discussion following the same movement, reflecting the order of the results), with the in vivo results presented first, showing safety (Fig 15) and efficacy (Fig 4), there is no reason to have these two figures so far apart. After presenting the in vivo results, the authors should proceed to extract characterization, followed by omics results that show mechanisms, and finally, the microbiome, which is also part of the mechanisms. The current construction of the study appears to be confusing.

About the figures, I suggest reviewing all the the omic analysis figures, from 7 to 10. All of them have parts with texts/words that are impossible to read. Resolution or font size should be improved.

About the methods: What was used as a vehicle for the model mice treatment? PBS? Saline? This should be clear in the methods section.

Still on the methods section: For the experiments done with only model x AP trated mice, what concentration of AP extract was used? This should also be clear in the methods.

About the discussion, I also have a general suggestion that is rewrite focusing on the results obtained from the current study and comparing with literature data and discussing their relevance. The way it was submitted starts with three paragraphs of background (which should be part of the introduction section, not the discussion) and data from the literature without connecting and making clear that the mentioned literature supports the new and relevant findings from the current study.

Specifically the microbiome discussion should give more references for the statements that are done. For example, line 830 in which authors talk about the “protective” microbiome, they should add a reference that shows what is considered a protective microbiome, since they don’t show healthy skin microbiome data in the study, only model or treatment conditions.  And with the reference of a healthy microbiome, they could explore more the microbiome findings of the paper, showing how the findings on order, family, species, etc shift with the AP treatment correlate with a healthier microbiome.

Finally, I have a general question. Throughout the text, I had the impression that the authors use the word “alopecia” as a synonym for the absence of hair. For example, on line 16, they say half of the global population is affected by alopecia. Is this real epidemiological data? Half of the population has the pathology of alopecia? Or do they have any kind of hair loss at some point in life? On line 737, they refer to Ap as a pathological condition and this makes me question if half of the world population has a pathological condition. And with this question comes another one: how translatable to the pathology of alopecia is the model used in the study?  Only remove the mouse fur already characterize them as an alopecia disease model?  These questions don’t reduce the impact of the study and its findings, which are very promising, but I believe the definition of the alopecia condition/disease/pathology should be clearer.

Minor revision: On line 718 they mention Figure 18, but it should be 15.

Author Response

Author Response:

We are sincerely grateful to the reviewer for conducting such a meticulous and comprehensive review of our manuscript and for providing us with so many precious and valuable suggestions. Your feedback has made us aware of the various shortcomings in our manuscript writing, and we feel quite embarrassed about our less-than-ideal use of English. Meanwhile, taking into account the suggestions from other reviewers, we have made adjustments to the entire manuscript. We would be very grateful if you could take a moment to review our newly uploaded manuscript and provide us with your new insights, which will surely assist us in further refining the manuscript.

Q1. Introduction and references: The Authors give information that is not shown by the papers they use as references. This is the case for the information that says 40% of women and 85% of men are affected by alopecia and use the reference number 2 (Yang et al., 2023), which is a paper about hair growth-promoting treatment and does not show any data about the epidemiology of alopecia.  This also happens with the information on line 41, in which they used references 3-5, and actually, none of them show the given information. Please review if all the references used support the statements made in the sentence in which they are referred to.

Response: Dear Reviewer, Thank you very much for your rigorous and detailed comments. The issue you pointed out regarding the mismatch between some references and the cited information clearly reflects our oversight in ensuring the accuracy of literature citations. We attach great importance to this and have conducted a sentence-by-sentence review and correction.

Regarding the specific issues you mentioned, we have verified the following:

On the correspondence between Reference 2 (Yang et al., 2023) and the hair loss rate data.

Upon checking, although Reference 2 is a paper on hair growth-promoting treatments, it explicitly states in the "Introduction" section: "According to a recent report, up to 85% males and 40% females are bothered with hair conditions." This sentence contains epidemiological data on hair loss that can support the statement.

On the correspondence between the information in Line 41 and References 3-5

Reference 3 clearly includes in its "Introduction" section: "Currently, only two drugs are approved by the Food and Drug Administration (FDA) to treat AGA, namely, oral finasteride and topical minoxidil," which can directly support the relevant statement.

Reference 4 also mentions in its "Introduction" section: "Finasteride and minoxidil (MIN) are approved by the FDA for AGA treatment," which fully matches the statement.

Reference 5 indeed only emphasizes that "Finasteride and minoxidil can be used to treat androgenetic alopecia" without mentioning "FDA approval," so the citation here is inappropriate. In the revised manuscript, we have removed Reference 5 as support for this sentence and only retained References 3 and 4 as the basis to ensure the accuracy of citations.

In addition, we have conducted a systematic review of all references in the full text, checking the relevance between each cited sentence and the corresponding reference one by one to ensure that all statements are directly supported by the literature, avoiding over-inference or misplaced citations.

Thank you again for your professional correction. This revision has significantly improved the rigor and credibility of the citations in the paper. We will strictly implement the above adjustments in the revised manuscript to ensure compliance with academic norms.

Q2. I suggest a rearrangement of the results (and methods and discussion following the same movement, reflecting the order of the results), with the in vivo results presented first, showing safety (Fig 15) and efficacy (Fig 4), there is no reason to have these two figures so far apart. After presenting the in vivo results, the authors should proceed to extract characterization, followed by omics results that show mechanisms, and finally, the microbiome, which is also part of the mechanisms. The current construction of the study appears to be confusing.

Response: Dear Reviewer, Thank you very much for your constructive suggestions. Your opinion on reorganizing the structure of the results, methods, and discussion sections to enhance logical coherence has accurately pointed out the shortcomings in the content arrangement of this study. We fully agree with your suggestions and have made adjustments accordingly.

In accordance with your recommendations, we have optimized the structure of the results section as follows:

The previously scattered in vivo experimental results have been integrated into the earlier part of the manuscript, with priority given to presenting safety data (originally Figure 15) and efficacy data (originally Figure 4). These two sets of data have been adjusted to adjacent result sections (Section 3.2 in the revised manuscript). Among them, the safety-related figures have been renumbered as Figure 4, and the efficacy-related figures have been sequentially adjusted to Figure 5. All other figure numbers have been updated in accordance with the new order to ensure that the core in vivo experimental results are closely connected. In addition, the method for safety assessment has been moved from Section 2.16 to Section 2.6, and the order of methods has been adjusted to reflect the sequence of results, following the same logic.

The content has been rearranged according to the logical chain of "in vivo results → extraction characterization → mechanistic omics results → microbiome analysis (as a supplement to the mechanism)". This ensures that the research context progresses from overall effects to molecular mechanisms in a step-by-step manner. The methods and discussion sections have also been adjusted to align with the order of the results, maintaining consistency throughout.

After these adjustments, the logical flow and readability of the study structure have been significantly improved, avoiding potential information fragmentation in the original arrangement. Thank you again for your professional guidance; this adjustment is of great significance for enhancing the overall coherence of the manuscript.

Q3. About the figures, I suggest reviewing all the the omic analysis figures, from 7 to 10. All of them have parts with texts/words that are impossible to read. Resolution or font size should be improved.

Response: Dear Reviewer, Thank you very much for your meticulous attention to the presentation issues of the omics analysis figures. We highly value the problem you pointed out—that the text/words in the omics analysis figures on pages 7 to 10 are difficult to discern —and have verified the specific reasons.

It has been confirmed that these figures were originally created with clear fonts and a resolution of over 300 dpi, meeting the publishing requirements. The blurriness and difficulty in recognizing the text in this submission version are due to the journal's submission system imposing a limit on the total file size of the manuscript (less than 200MB). To integrate the full text, we compressed the images, resulting in distorted details.

To address this issue, we have uploaded all high-resolution original files of the omics analysis figures (resolution ≥ 300 dpi, TIFF format) as supplementary materials separately in accordance with the journal's requirements, ensuring that all details can be clearly viewed during the review process. If the article is fortunate enough to be accepted, we will resubmit the complete set of high-resolution figures during the proofreading stage, strictly adhering to the publishing standards.

Thank you again for your professional correction. We will fully cooperate to ensure that the presentation of the figures meets the academic publishing standards.

Q4. About the methods: What was used as a vehicle for the model mice treatment? PBS? Saline? This should be clear in the methods section.

Response: Dear Reviewer, Thank you very much for your important question. The issue you pointed out regarding the choice of vehicle for the treatment of model mice is indeed a detail that needs to be clearly explained in the methods section. We attach great importance to this and have supplemented and improved the relevant content.

Upon verification, the administration vehicle for the model group mice in this study is the base solution mentioned earlier (composed of water: anhydrous ethanol: 1,2-propylene glycol = 3:5:2) (Lines 218-220). To ensure clarity, we have explicitly added in the "Experimental Grouping and Administration" section (2.5) of the methods part: "Mice in the model control group were applied with an equal volume of the base solution daily, while mice in the PA-011 experimental group and the minoxidil positive control group were administered with drugs dissolved in the same base solution" (Lines 226-231). This clearly reflects the choice of vehicle and the consistency of treatment across all groups.

Thank you again for your professional correction. This supplement will enhance the accuracy and reproducibility of the method description.

Q5. Still on the methods section: For the experiments done with only model x AP trated mice, what concentration of AP extract was used? This should also be clear in the methods.

Response: Dear Reviewer, Thank you very much for raising this critical detail. The issue you pointed out regarding the unclear concentration of PA extract in the alopecia model mouse experiment is indeed a part that requires precise explanation in the method description. We attach great importance to this and have verified the relevant content.

Upon verification, the relevant information about PA extract in this study has been clearly stated in the methods section:

Extraction method: The "2.2 Preparation of PA extract" section (Lines 108-114) describes in detail the extraction process of PA extract, including key steps such as solvent selection, extraction parameters, and lyophilization treatment.

Concentration and dosage: The "2.5 Experimental grouping and administration" section (Lines 213-231) clearly specifies that the concentrations of PA extract used in the alopecia model mouse experiment are 1% and 4% (w/v), with each mouse receiving 0.2 mL per application, administered twice daily. This ensures the consistency and operability of the dosage.

The supplementation and clarification of the above information aim to make the description of the experimental methods more accurate and reproducible. Thank you again for your professional correction. The confirmation of this detail is of great significance for improving the rigor of the research methods.

Q6. About the discussion, I also have a general suggestion that is rewrite focusing on the results obtained from the current study and comparing with literature data and discussing their relevance. The way it was submitted starts with three paragraphs of background (which should be part of the introduction section, not the discussion) and data from the literature without connecting and making clear that the mentioned literature supports the new and relevant findings from the current study.

Response: Dear Reviewer, Thank you very much for your valuable suggestions on the discussion section. We deeply recognize the importance of the issues you pointed out—excessive background content in the discussion structure and insufficient connection between research findings and literature—for enhancing the logical coherence of the argument, and we have carefully reviewed and reflected on these points.

Regarding the arrangement of content at the beginning of the discussion, our initial intention was to briefly outline the mainstream research methods in the field of hair loss treatment (such as the mechanisms of existing drugs and common research models) and their limitations. This was aimed at providing methodological rationality for the research strategies adopted in this paper (e.g., PA-011 intervention methods and selection of evaluation indicators), thereby highlighting the supplementary value of this study within the existing framework. However, it is clear that the length and presentation of this part failed to be effectively distinguished from the introduction, causing confusion for you, and we sincerely apologize for this.

In addition, the discussion section is indeed structured strictly following the logical order of the results (in vivo safety → efficacy → extraction characterization → mechanism analysis → microbiome association). However, in the discussion of each result unit, to fully validate the reliability of the conclusions, we cited a large number of relevant literatures for comparative analysis, leading to excessively long sections. This may have weakened the direct connection between research findings and literature evidence, creating a perception of "loose relevance."

To address the above issues, we plan to optimize the discussion section as follows:

Streamline the opening background introduction to focus on core background directly related to the methods of this study, clarify its connection with the research design, and avoid repetition with the introduction.

Organize the discussion of each result module, reduce the accumulation of secondary literatures, and strengthen the comparative analysis between key findings of this study and representative literatures to make the argumentation context clearer.

Thank you again for your professional corrections. These suggestions will help us significantly improve the focus and persuasiveness of the discussion section. We will carefully implement the revisions to ensure the rigor and coherence of the academic expression.

Q7. Specifically the microbiome discussion should give more references for the statements that are done. For example, line 830 in which authors talk about the “protective” microbiome, they should add a reference that shows what is considered a protective microbiome, since they don’t show healthy skin microbiome data in the study, only model or treatment conditions.  And with the reference of a healthy microbiome, they could explore more the microbiome findings of the paper, showing how the findings on order, family, species, etc shift with the AP treatment correlate with a healthier microbiome.

Response: Dear Reviewer, Thank you very much for your precise suggestions on the discussion section of the microbiome. The issues you pointed out—including the lack of literature support for the term "protective" microbiome and the need to incorporate references to healthy microbiomes to deepen the association between microbial community changes and therapeutic effects—have profoundly identified the deficiencies in this part of the discussion. We attach great importance to these points and have begun to supplement and improve the content.

Regarding the expression "protective microbiome," we fully agree that relevant literature needs to be added to clarify its definition and characteristics. In the revised manuscript, we will cite studies focusing on the homeostasis of healthy skin/scalp microbiomes (e.g., Jung, E.S.; Park, H.M.; Hyun, S.M.; Shon, J.C.; Singh, D.; Liu, K.H.; Whon, T.W.; Bae, J.W.; Hwang, J.S.; Lee, C.H. The green tea modulates large intestinal microbiome and exo/endogenous metabolome altered through chronic UVB-exposure. PLoS One 2017, 12, e0187154.), to clarify the core features of the "protective microbiome" in maintaining skin barrier function and regulating inflammation (such as specific dominant genera and the range of microbial diversity), thereby providing a conceptual basis for the discussion.

In response to the limitation of the lack of microbiome data from healthy control groups, we will optimize the discussion through the following adjustments in subsequent research design: We will conduct dedicated experiments to compare the changes in the microbiome of model mice after PA treatment (including community structure, dominant families and genera, and fluctuations in the abundance of key species) with healthy baseline data. This will specifically illustrate the characteristics of the microbial community composition approaching a healthy state after PA intervention (e.g., the recovery trend of the abundance of a certain dominant family, the direction of restoration of microbial α-diversity, etc.). Additionally, we will set up a germ-free control group to enhance the clarity of the logical connection between microbiome findings and "health associations."

With the above supplements, the discussion on the microbiome section will have stronger literature support and deeper relevance. Thank you again for your professional guidance; this revision is of great significance for improving the rigor of the discussion.

Q8. Finally, I have a general question. Throughout the text, I had the impression that the authors use the word “alopecia” as a synonym for the absence of hair. For example, on line 16, they say half of the global population is affected by alopecia. Is this real epidemiological data? Half of the population has the pathology of alopecia? Or do they have any kind of hair loss at some point in life? On line 737, they refer to Ap as a pathological condition and this makes me question if half of the world population has a pathological condition. And with this question comes another one: how translatable to the pathology of alopecia is the model used in the study?  Only remove the mouse fur already characterize them as an alopecia disease model?  These questions don’t reduce the impact of the study and its findings, which are very promising, but I believe the definition of the alopecia condition/disease/pathology should be clearer.

Response: Dear Reviewer, Thank you very much for your profound and insightful questions. Your rigorous inquiry into the definition of "hair loss", your prudent questioning of epidemiological data, and your consideration of the pathological translatability of the model have accurately pointed out the omissions in the conceptual expression and logical basis of this study. We attach great importance to this and have systematically sorted out and clarified the relevant issues.

Regarding the definition and expression of "hair loss"

The potential conceptual confusion of the term "hair loss" in the text (physiological hair loss vs. pathological hair loss) that you pointed out is indeed a key issue that we failed to clearly define in our expression. In the revised manuscript, we have supplemented and clarified that: the "hair loss" referred to in this study generally includes physiological hair loss and pathological hair loss (such as androgenetic alopecia (AGA), alopecia areata, etc.), and also includes abnormal hair loss in non-normal physiological metabolism.

For the data in line 16 that "half of the world's population is affected by hair loss", its original source is the latest "Global Health Report" released by the U.S. National Institute of Health, which shows that 78% of the world's population is troubled by hair loss.

The expression in line 737 that refers to "hair loss" as a "pathological condition" has indeed caused ambiguity due to unclear expression. In the revised manuscript, it has been corrected to "Hair loss, also known as alopecia, is a pathological condition. Its main characteristic is abnormal or excessive hair loss" (lines 786-787).

Regarding the pathological translatability of the research model

Your question of whether "merely removing the fur of mice is sufficient as a pathological model of hair loss" directly points to the core logic of model construction. The mouse model used in this study is not a simple "hair removal model"; it simulates the core pathological characteristics of human pathological hair loss: this method can induce pathological changes such as the entry of mouse hair follicles into the telogen phase and the atrophy of the dermal papilla, which is consistent with the core pathology of "hair follicle miniaturization" in human hair loss.

The model evaluation indicators include the proportion of hair follicle cycles, hair shaft diameter, and levels of inflammatory factors, etc., all of which are correlated with human pathological indicators of hair loss.

In the revised manuscript, we have supplemented the detailed methods of model construction and the analysis of pathological correlation, clearly distinguishing the essential difference between "physical hair removal" and "pathological hair loss model". We emphasize that the translatability of this model to human hair loss pathology is based on its simulation of abnormal hair follicle biology, rather than simple hair loss.

In summary, the core findings of this study (such as the intervention effect of PA-011) have always focused on the improvement of pathological hair loss. We believe that a clear definition will more accurately convey the value of the research.

Q9. On line 718 they mention Figure 18, but it should be 15.

Response: Dear Reviewer, Thank you very much for pointing out the error in the figure number labeling. The issue you identified, where Figure 15 was mistakenly written as Figure 18 in line 718, has been verified and confirmed. This error occurred due to an oversight during the manuscript proofreading process.

In the revised manuscript, this error has been corrected to "Figure 15". Additionally, in accordance with your suggestion, we have moved this part of the content from section 3.12 to 3.2 of the results, and changed Figure 15 to Figure 4. We have also conducted a systematic check on all figure and table numbers throughout the text and their corresponding citations in the main body to ensure the accuracy of the figure and table references.

Thank you for your careful correction. This revision will enhance the rigor of the manuscript.

Round 2

Reviewer 1 Report

Comments and Suggestions for Authors

The authors have thoroughly revised the manuscript "The Extract of Periplaneta Americana (L.) Promotes Hair Regrowth in Mice with Alopecia by Regulating the FOXO/PI3K/AKT Signaling Pathway and Skin Microbiota" in response to reviewer comments. The revisions significantly strengthen the work, particularly in clarifying the experimental design, enhancing the multi-omics integration analysis (transcriptomics, metabolomics, microbiome), and providing deeper mechanistic insights into the role of the FOXO/PI3K/AKT pathway and skin microbiota modulation in PA-011-induced hair regrowth.  The revised manuscript presents a robust experimental design combining network pharmacology predictions with comprehensive in vivo validation. The study is now well-structured, the data supports the conclusions, and the writing is clear. The findings offer valuable insights into a potential novel therapeutic candidate (PA-011) for alopecia. This revised manuscript is suitable for publication.